# Graphon-Explainer: Generating Model-Level Explanations for Graph Neural Networks using Graphons

**Sayan Saha**  *sayansaha181196@gmail.com*
*Machine Intelligence Unit*
*Indian Statistical Institute, Kolkata*

**Sanghamitra Bandyopadhyay**  *sanghami@isical.ac.in*
*Machine Intelligence Unit*
*Indian Statistical Institute, Kolkata*

**Reviewed on OpenReview:** *https://openreview.net/forum?id=yHUtuvoIQv*

## Abstract

Graph Neural Networks (GNNs) form the backbone of several state-of-the-art methods for performing machine learning tasks on graphs. As GNNs find application across diverse real-world scenarios, ensuring their interpretability and reliability becomes imperative. In this paper, we propose Graphon-Explainer, a model-level explanation method to elucidate the high-level decision-making process of a GNN. Graphon-Explainer learns a graphon—a symmetric, continuous function viewed as a weighted adjacency matrix of an infinitely large graph—to approximate the distribution of a target class as learned by the GNN. The learned graphon then acts as a generative model, yielding distinct graph motifs deemed significant by the GNN for the target class. Unlike existing model-level explanation methods for GNNs, which are limited to explaining a GNN for individual target classes, Graphon-Explainer can also generate synthetic graphs close to the decision boundary between two target classes by interpolating graphons of both classes, aiding in characterizing the GNN model's decision boundary. Furthermore, Graphon-Explainer is model-agnostic, does not rely on additional black-box models, and does not require manually specified handcrafted constraints for explanation generation. The effectiveness of our method is validated through thorough theoretical analysis and extensive experimentation on both synthetic and real-world datasets on the task of graph classification. Results demonstrate its capability to effectively learn and generate diverse graph patterns identified by a trained GNN, thus enhancing its interpretability for end-users.

## 1 Introduction

Graph Neural Networks(GNNs)(Zhang et al., 2020; Wu et al., 2020) have attained state-of-the-art performance on various graph tasks such as node classification, graph classification and link prediction. They are increasingly being deployed in the real world in diverse applications such as drug discovery and molecular structure prediction(Merchant et al., 2023), weather forecasting(Lam et al., 2023) and recommender systems(Wu et al., 2022). The integration of GNNs into real-world applications comes with the dire need to make them interpretable and trustworthy for stakeholders who regularly interact with or rely on such models. Although substantial research has been conducted on explainability in domains like vision and text analysis (Kim et al., 2018; Selvaraju et al., 2017; Ribeiro et al., 2016; Chen et al., 2018), these methods are challenging to directly apply to graphs. This difficulty stems from the discrete nature of graph data, which poses challenges for optimization using gradient-based techniques. Moreover, unlike images, graph data exhibits irregularity as each graph may contain a varying number of nodes, and its validity often hinges upon adhering to domain specific structural constraints.

Existing post hoc explainability methods for GNNs can be categorized into two classes(Yuan et al., 2022): Instance-Level methods and Model-level methods. Instance-level methods offer explanations for the model's predictions on individual inputs, while model-level methods investigate the general behavior of the model without reference to specific inputs. When the aim is to evaluate the reliability and trustworthiness of a model, one needs to examine instance-level explanations across several instances to draw a rigorous conclusion about the behavior of the model. Therefore, model-level explanations are more apt in such contexts. Furthermore, it has been demonstrated in Faber et al. (2021) that instance-level explanations fail to produce faithful explanations when the GNN model is trained on a binary classification problem and suffers from the bias-attribution issue. Model-level explanations can not only produce faithful explanations in this scenario but also diagnose the bias attribution issue. Although model-level explanation methods offer substantial benefits over instance-level approaches, they have received comparatively less attention in the literature. Current methods such as XGNN (Yuan et al., 2020), D4Explainer (Chen et al., 2024), and GNNInterpreter (Wang & Shen, 2022) focus on providing explanations for specific target classes but fall short in elucidating the decision boundary of the underlying model. On the other hand, GNNBoundary (Wang & Shen, 2024) can generate graphs near the decision boundary between two target classes but lacks class-specific explanations.

In this paper, we introduce Graphon-Explainer, a holistic model-level explanation method that can not only generate explanations for individual target classes but also produce synthetic motifs that help characterize the decision boundaries between two classes. To the best of our knowledge, this is the first method capable of offering both class-specific explanations and insights into the decision boundary of the target classes. Graphon-Explainer approximates the distribution of graphs classified to a particular target class by the GNN model by estimating a graphon function for that target class. The learned graphon serves as a generator, producing graph motifs that capture the distinctive features learned by the GNN about the target class. Unlike graphs, graphons possess a regular structure and alignment, facilitating the interpolation of graphons from two target classes to produce a graphon that can generate synthetic graphs lying close to the decision boundary between the two classes. Graphon-Explainer has several advantages over current state-of-the-art model-level methods. Unlike, XGNN(Yuan et al., 2020) and D4Explainer(Chen et al., 2024) it does not need another black box deep learning model to generate explanations. Similarly, unlike GNNInterpreter (Wang & Shen, 2022), which requires access to training data embeddings to generate explanations, Graphon-Explainer only requires access to the embeddings of the query dataset that the user possesses, while seeking to explain the model's behavior.

## 2 Preliminaries

We introduce some graphon related mathematical concepts in this section that we use later for the theoretical analysis.

**Graphon:** A graphon $S : [0,1]^2 \to [0,1]$ is a symmetric, continuous, and measurable function which can be viewed as a weighted adjacency matrix of a graph with infinitely many nodes. Given two points $x_i, x_j \in [0,1]$, $S(i,j)$ can be interpreted as representing an weighted edge between nodes $i$ and $j$. Graphons arise as limits of sequences of dense graphs sharing a common set of topological attributes in the graph theory literature and can serve as a generator of graphs having similar topological properties. For instance, sampling $n$ points uniformly from the unit interval and setting $S(x,y) = p \quad \forall x,y \in [0,1]$, where $p \in [0,1]$ is a constant generates a Erdős-Rényi random graph $G(n,p)$.

**Cut Norm:** The cut norm of a graphon $S$ can be defined as:

$$\|S\|_\square = \sup_{X,Y \subset [0,1]} \left| \int_{X \times Y} S(x,y) \right| \tag{1}$$

For two graphons,$S^1$ and $S^2$, a measure of distance between them can be defined using the cut norm $\|S^1 - S^2\|_\square$. Lower values of the cut norm indicate that the graphons $S^1$ and $S^2$ have a high degree of structural similarity. Graphon estimation methods frequently use step functions as it has been proven (Lovász, 2012) that graphons can always be approximated with arbitrary accuracy in the cut norm using step functions.

**Graph Homomorphism:** Given two graphs $F$ and $G$, let their node sets be denoted as $V(F)$ and

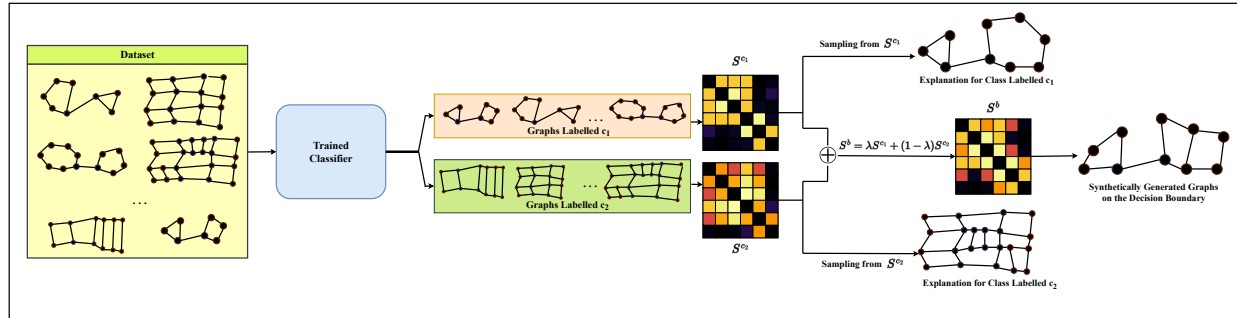

Figure 1: A demonstration of the Graphon-Explainer workflow on a binary classification problem. First, a trained classifier categorizes a dataset of graphs into two classes $c_1$ and $c_2$. Then, a graphon is estimated for each class ($S^{c_1}$ and $S^{c_2}$) using the graphs assigned to that class. These estimated graphons are sampled to generate explanations for both classes by optimizing their class scores, with the classifier evaluating the explanations. Additionally, the estimated graphons are interpolated to create a new graphon $S^b$, which is sampled to produce graphs that lie near the decision boundary of both the classes. These graphs exhibit a mixture of motifs from both classes.

$V(G)$ respectively and their edge sets be denoted as $E(F)$ and $E(G)$ respectively. A graph homomorphism $\psi$ is a map from $V(F) \rightarrow V(G)$ such that if $(u,v) \in E(F)$ then $(\psi(u), \psi(v)) \in E(G)$. Let, $hom(F,G)$ denote the total number of graph homomorphisms from $F$ to $G$. Then, the homomorphism density $t(F,G)$ can be defined as $t(F,G) = \frac{hom(F,G)}{|V(G)|^{|V(F)|}}$. Homomorphism density measures the fraction of structure-preserving maps from $F$ to $G$. The notion of homomorphism density can be naturally extended to graphons. Given a graph $F$, its homomorphism density with respect to a graphon $S$ can be defined as $t(F,S) = \int_{[0,1]^{V(F)}} \prod_{i,j \in E(F)} S(x_i, x_j) \prod_{i \in V(F)} dx_i$.

## 3 Related Work

**Graph Neural Networks:** GNNs work by learning a node's representation by a message passing mechanism. Each node's representation is learned by aggregating features of neighboring nodes and combining them with its own features. Different architectures (Veličković et al., 2018; Kipf & Welling, 2016; Gilmer et al., 2017) differ in their aggregation functions and message passing functions but the learning strategy remains similar across architectures.

**Instance-Level Explanations:** Instance level methods provide an explanation corresponding to each input. Typically, they aim to identify salient subgraphs, nodes or edges that is deemed important by the GNN for the graph to be classified into a particular class. According to the survey (Kakkad et al., 2023) they can be categorized into: decomposition based methods(Pope et al., 2019; Feng et al., 2023), gradient based methods(Baldassarre & Azizpour, 2019; Huang et al., 2022), surrogate methods(Zhang et al., 2021; Vu & Thai, 2020) and perturbation based methods(Schlichtkrull et al., 2020; Yuan et al., 2021; Lucic et al., 2022; Lin et al., 2022). Sharing a similar goal as our method, MotifExplainer(Yu & Gao, 2023) employs domain specific motif extraction rules to identify a candidate set of explanation motifs and uses an attention based mechanism to identify motifs from the candidate set in an instance graph that is important for the categorization of the graph to a particular class by the GNN classifier. However, our approach differs significantly in two key ways. First, while MotifExplainer relies on domain-specific rules for motif extraction, our method does not. Second, MotifExplainer provides instance-level explanations, which are not generalizable across different graphs. In contrast, our model-level explanation method generates motifs that reflect the features the GNN has learned for an entire target class, rather than focusing on individual instance graphs.

**Model-Level Explanations:** The goal of model-level explainability is to understand a model's general behavior by generating graph patterns that trigger specific predictions. Different model-level methods differ in their graph generation methods. XGNN(Yuan et al., 2020) trains a reinforcement learning model using the policy gradient method to maximize a reward function designed using handcrafted constraints to generate graphs that serve as explanations of a target class. D4Explainer(Chen et al., 2024) generates graphs

by training a diffusion model which requires significant computational resources and time to generate an explanation. GNNInterpreter(Wang & Shen, 2022) was the first model-level method that did not rely on another black-box neural network architecture to generate explanations. However, during training, it requires the embeddings of the GNN's training data as input for an objective function it aims to maximize, necessitating access to both the training data and the hidden layers of the GNN. It is worthwhile to mention here that Graphon-Explainer does not require access to the GNN's internal components, does not use hand-crafted constraints or employ another black-box model to generate explanations. Another recent method GNNBoundary(Wang & Shen, 2024) is a model-level explanation method that illustrates the decision making process of a GNN by generating graphs that lie close to the decision boundary between two classes, however, it cannot produce explanations for individual target classes. Other notable works in model-level explainability include GDM (Nian et al., 2024), which generates model-level explanations by minimizing the distance between its explanations and training graph embeddings during the GNN's training process, and MAGE (Yu & Gao, 2024), which provides motif-based model-level explanations for classification tasks on molecular datasets. However, it is important to note that these methods also share the limitation of producing explanations only for individual target classes, without illustrating the decision boundary.

**Other Methods of GNN Explanations:** There are also GNN explanation methods that do not strictly belong to either the instance-level or model-level categories, and some approaches combine both. A notable example of the former is GraphChef (Müller et al., 2024), which develops an intrinsically explainable GNN capable of explaining its decision-making process on any dataset by generating a decision tree. An example of the latter category (Azzolin et al., 2022) uses instance-level explanations as input and combines them using Boolean operations to construct model-level explanations.

**Graphons:** Graphons have been extensively studied as limits of graph sequences in the literature (Lovász & Szegedy, 2006; Lovász, 2012). They have also been explored in the contexts of Graph Neural Networks (GNNs) (Han et al., 2022; Ruiz et al., 2020) and network theory (Vizuete et al., 2021). Approaches to approximating graphons can be categorized into two main lines of work. The first utilizes stochastic block models, including methods like stochastic block model approximation (SBA)(Airoldi et al., 2013) and sorting and smoothing (SAS)(Chan & Airoldi, 2014). The second line employs low-rank matrix approximations, such as matrix completion(Keshavan et al., 2010) and universal singular value thresholding (USVT)(Chatterjee, 2015).

## 4 Method

---

**Algorithm 1** Generating Explanations using Graphon-Explainer

---

**Hyperparameters:**
- $n_{samples}$: Number of samples to be generated by the estimated generative model.
- $K$: Number of nodes in the generated explanation
- $\lambda$: Weight of a graphon during linear interpolation of graphons of two target classes

**Input:** Trained GNN graph classifier model $f(\cdot)$, Sets of graphs $\{D_{c_1}, \cdots, Dc_k\}$ labeled by $f$ as belonging to classes $\{c_1, \cdots, c_k\}$, Graphon Estimator $g$

**Procedure:**
- Generating explanations for class $c$:
    - Estimate graphon $S^{c_i}$ for class $c_i$ on $K$ partitions and USVT graphon extimator using Algorithm 4.
    - Sample $n_{samples}$ from the estimated graphon using as described in Algorithm 4.
    - Select the sample having the highest target class score as the explanation.
- Generating graphs close to the decision boundary of two classes $c_1$ and $c_2$:
    - Combine estimated graphons $S^{c_1}$ and $S^{c_2}$ linearly with weights $\lambda$ to get the boundary graphon $S^b = \lambda S^{c_1} + (1 - \lambda)S^{c_2}$.
    - Sample $n_{samples}$ from $S^b$ using equation 4.
    - Select the sample that minimizes equation 3.

---

### 4.1 Problem Setup:

The goal of model-level explanation is to pinpoint discriminative graph patterns that trigger a specific class prediction from a trained graph classification model $f(.)$. Formally, the goal of producing model level explanation for a target class $c$ can be stated as identifying $G^*$ for which,

$$G^* = \underset{G}{\text{argmax}} \ P(f(G) = c) \tag{2}$$

Assuming, we have an unknown joint distribution of graphs and class labels $P(G, C)$, the optimization takes place over the samples drawn from the class conditional distribution $P(G|C = c)$ which we assume in line with prior works (Wang & Shen, 2022; 2024) can be written as $P(X|C = c)P(A|C = c)$, the product of the distribution of node feature matrices and adjacency matrices. In this work, the distribution of the adjacency matrices is approximated by the estimated graphon and the distribution of the node feature matrices is approximated by a probabilistic weighting mechanism on the pool of node features of graphs that are classfied to a particular target class by $f(.)$. It should be noted that there might exist multiple distinct motifs $G^*$ that maximize the equation 2. Such motifs shed light on the graph patterns learnt by the model $f(.)$ for the particular target class.

Beyond the generation of explanations for individual target classes, Graphon-Explainer can also generate graphs that lie close to the decision boundary of two target classes. Formally stated, for a pair of classes $c_1$, $c_2$, the objective is to find motifs $G_b$ such that

$$G_b = \underset{G}{\text{argmin}} \ |P(f_{c_1}(G)) - P(f_{c_2}(G))| \quad \text{subject to} \quad P(f_{c_i}(G)) > P(f_{c_j}(G)) \quad \forall i \in \{1, 2\}, \forall j \notin 1, 2 \tag{3}$$

$P(f_{c_i}(G))$ denotes the probability assigned by $f(.)$ to $G$ for class $c_i$. Minimizing this objective ensures that $G_b$ lies close to the decision boundary of both classes.

### 4.2 Implementation Setting

Graphon-Explainer is a model-agnostic method that can be used to explain any GNN classifier model. It does not need access to hidden layers of the GNN classifier or knowledge of the process by which the GNN classifier outputs a label for an input graph. Graphon-Explainer only assumes the ability to query the trained GNN classifier for predictions. Figure 1 illustrates the workflow of Graphon-Explainer, which we will explain in detail below.

We begin with a dataset of graphs, represented as $D$, wherein each graph is associated with a distinct class label, denoted by $c$, with $c$ taking on values within the set $\{1, \cdots, C\}$. Additionally, a trained GNN graph classifier, represented as $f(.)$, can be queried to predict the class of a given graph within this dataset. Consequently, when presented with a collection of graphs $D_c \subseteq D$, identified by $f(.)$ as belonging to a specific target class $c$, Graphon-Explainer proceeds to generate explanations for the class $c$. Graphon-Explainer generates explanations in two steps. First, it estimates a graphon for the target class $c$ using the graphs in $D_c$. Second, it samples from the estimated graphon while optimizing the objective in Equation 2 to generate graphs that exhibit the discriminative motifs characteristic of class $c$ as identified by $f(.)$. Further, when provided with datasets $D_{c_1}$ and $D_{c_2}$ of graphs that belong to two target classes $c1$ and $c_2$ according to $f(.)$, Graphon-Explainer can generate examples that lie close to the decision boundary of the two classes. Generating examples that lie on the decision boundary involves estimating the graphon for each class from the corresponding datasets as a first step. This step is followed by taking a convex combination of the graphons of both these classes to estimate the graphon of graphs that lie on the boundary. Finally, sampling from this graphon while optimizing the objective in Equation 3 yields examples that lie on the decision boundary. We contain a combination of discriminative motifs present in classes $c_1$ and $c_2$.

### 4.3 Graphon Estimation

The first step in generating explanations using Graphon-Explainer involves approximating graphons of each target class using graphs classified into a target class by the trained classifier $f$. Estimating a graphon for a

class of real-world graphs is inherently difficult as it is an unknown function lacking a closed-form expression. Therefore, in line with previous research (Xu et al., 2021; Han et al., 2022; Airoldi et al., 2013) we employ a step function method for graphon estimation. Let, $P = \{P_1, \cdots, P_K\}$ be a partition of $[0,1]$ into $K$ disjoint intervals. A step function $S_P : [0,1]^2 \to [0,1]$ can be defined as $S_P(x,y) = \sum_{k,k'=1}^{K} s_{kk'}(x,y)\mathbb{I}_{P_k \times P_{k'}}$, where each $s_{kk'} \in [0,1]$ and $\mathbb{I}$ is an indicator function which is equal to 1 when $(x,y) \in P_k \times P_{k'}$ otherwise it is equal to 0. Hence, the step function $S_P$ over $K$ partitions of the unit interval can also be seen as a matrix $S_P = [s_{kk'}] \in [0,1]^{K \times K}$. For our purposes, we set $K$ to be equal to the median number of nodes in the dataset $D$ across all target classes. We use the USVT method (Chatterjee, 2015) to estimate the step function. This method aligns the nodes of the graph in each dataset based on certain node properties and then proceeds to estimate the step function from the aligned adjacency matrices. The weak regularity lemma of graphons (Lovász & Szegedy, 2006) guarantees that any graphon can be accurately approximated well in the cut norm by step functions. Algorithm 4 describes the method we use to approximate graphons. The algorithm and the lemma are detailed in Appendix C.

### 4.4 Generation of Explanation for a Target Class

Once, the graphon for the target class $c$ is approximated by the step function $S^c$ using graphs in $D_c$, one can sample from this graphon to generate explanations for the target class $c$. Given $S^c$, we can generate a $N$ node graph through the following two-step sampling procedure:

$$
\begin{aligned}
u_n &\sim \text{Uniform}([0,1]) \quad \forall n \in \{1, \cdots, N\} \\
a_{nn'} &\sim \text{Bern}(S^c(u_n, u_{n'})) \quad \forall n, n' \in \{1, \cdots, N\}
\end{aligned}
\tag{4}
$$

The first step samples $N$ nodes from a uniform distribution on the unit interval $[0,1]$. The second step generates an adjacency matrix $A = [a_{nn'}] \in \{0,1\}^{N \times N}$ by sampling each entry from a Bernoulli distribution with parameters defined using the estimated graphon $S^c$. Graphs sampled from $S^c$ using this sampling process would contain discriminative motifs that are significant for the classifier $f(.)$ to label the generated graphs as belonging to the target class $c$. The generation of node features of synthetic graphs is done in two phases. During the graphon estimation phase, we align the original node features with the adjacency matrices, resulting in a set of aligned node features for each graphon. We pool these aligned features and assign probabilistic weights to them depending on the their frequency of occurence. Node features are then sampled according to their probabilistic weight from the pool. The graph generation process is detailed in Algorithm 4.

### 4.5 Generation of Graphs near the Decision Boundary

Generation of synthetic examples lying on the decision boundary of two target classes requires estimation of the graphon functions of the corresponding target classes as a first step. Given two target classes $c_1$ and $c_2$ and their estimated graphons $S^{c_1}$ and $S^{c_2}$, we take a convex combination of these graphons : $S^b = \lambda S^{c_1} + (1-\lambda)S^{c_2}$. Interpolating graphons of two target classes yields a graphon which is a mixture of the graphons of both classes. Graphs can be sampled from $S^b$ through the same sampling procedure as shown in Equation 4. The node features of the boundary motifs are generated by first sampling the node features from each graphon, as outlined in Algorithm 4. Next, the sampled features in each node feature matrix are reweighted according to the mixup parameter of each graphon. Finally, each node of the generated graph is resampled using the corresponding weights as probabilities. Graphs sampled from the interpolated graphon partially contain discriminative features of both the classes which places them close to the decision boundary of the two target classes. In a multi-class classification problem, an evaluation is performed to determine if two classes share a decision boundary. This process begins by randomly sampling $N$ pairs of graphs from the two target classes. For each pair, the corresponding graph embeddings are extracted from the penultimate layer of the classifier, before the final linear layer that produces class-specific logits. Next, these embeddings are interpolated, and the classifier's output on the interpolated embeddings is analyzed to see if it crosses into the decision region of another class. A score is then calculated for each pair based on this analysis. If the cumulative score across the $K$ pairs exceeds a predefined threshold, the two target classes are considered adjacent, sharing a decision boundary. The pseudocode of this procedure is detailed

in Algorithm 5 in Appendix E. Synthetic graphs close to their decision boundary can then be obtained by sampling from their interpolated graphon of obtained by a mixup of graphons of both the target classes. Algorithm 1 describes the method to generate explanations and boundary motifs using Graphon-Explainer.

## 5 Theoretical Analysis

We give theoretical guarantees to demonstrate that graphs generated from a graphon that is a convex combination of the estimated graphons of two target classes would contain discriminative motifs belonging to both classes. We have shown (in Section 4.5 ) how such graphons can be used to generate synthetic graphs that lie close to the decision boundary of the two target classes. This validates the intuitive notion that graphs that lie on or close to the decision boundary of two target classes contain discriminative features of both classes. We first give a concrete definition of a discriminative motif and state a few basic assumptions on which the theoretical guarantees rely on.

**Definition** A discriminative motif $M_G$ of graph $G$ is the smallest subgraph of $G$ that gives $G$ its class identity. In, other words $M_G$ is a subgraph with a minimal number of nodes and edges, which is essential for $G$ to be identified as belonging to its corresponding class.

Note that, if there exists a graph homomorphism from a motif $M_G$ to a graph $G$, then the motif $M_G$ exists in graph $G$. Thus, the homomorphism density $t(M_G, G)$ gives a measure of the frequency of occurence of $M_G$ in $G$ We assume that every graph belonging to a target class $c$ contains a discriminative motif $M_c$. Further, we also assume that any collection of graphs $D_c$ that has been identified by a classifier as belonging to a class $c$ contains a finite set of discriminative motifs $\mathbb{M}_c$. The following theorem shows that the boundary graphon partially contains discriminative motifs belonging to both the target classes.

**Theorem 5.1.** *Let $D_{c_1}$ and $D_{c_2}$ be two collections of graphs belonging to the target classes $c_1$ and $c_2$, respectively, with their estimated graphons $S^{c_1}$ and $S^{c_2}$. Define the boundary graphon as $S^b = \lambda S^{c_1} + (1 - \lambda)S^{c_2}$. For any discriminative motif $M_{c_1} \in \mathbb{M}_{c_1}$ and $M_{c_2} \in \mathbb{M}_{c_2}$, the difference in the homomorphism density of $M_{c_1}$ and $M_{c_2}$ with the boundary graphon $S^b$ compared to their respective graphons $S^{c_1}$ and $S^{c_2}$ is bounded above by:*

$$|t(M_{c_1}, S^{c_1}) - t(M_{c_1}, S^b)| \leq (1 - \lambda)|E(M_{c_1})|\|S^{c_1} - S^{c_2}\|_\Box$$
$$|t(M_{c_2}, S^{c_2}) - t(M_{c_2}, S^b)| \leq \lambda|E(M_{c_2})|\|S^{c_1} - S^{c_2}\|_\Box$$

*Proof.* The proof is detailed in Appendix D.1. □

Theorem 5.1 establishes an upper bound on the difference in homomorphism densities between a discriminative motif of a target class and both the graphon of the target class and the boundary graphon. In other words, the difference between the frequency of occurence of a motif in the graphon of a target class and in the boundary graphon is upper bounded. This upper bound is solely dependent on the hyperparameter $\lambda$, since the values of $|E(M_{c_1})|$ and $|E(M_{c_2})|$ are constants determined by the discriminative motifs.

Next, we present a theoretical guarantee that graphs generated from the graphon $S^b$ will inherit the motifs contained in $S^b$.

**Theorem 5.2.** *Let $S^b$ be the boundary graphon and assume that it contains a discriminative motif $M_b$. Then any random graph $G$ on $n$ nodes generated from $S^b$ satisfies:*

$$P(|t(M_b, G) - t(F, S^b)| > \epsilon) \leq 2\exp\left(-\frac{\epsilon n^2}{18|V(M_b)|^2}\right)$$

*Proof.* The proof is detailed in Appendix D.3 □

Theorem 5.2 demonstrates that the topology of a random graph sampled from the boundary graphon closely resembles the topology of the boundary graphon itself, which includes discriminative motifs from both classes. Consequently, the theoretical analysis in this section confirms that graphs generated by the boundary graphon $S^b$ will contain discriminative motifs from both target classes. Further, a time complexity analysis and a theoretical guarantee of diversity in the generated explanations of Graphon-Explainer is provided in Appendix D.3 and D.2.

# 6 Experimental Analysis

We evaluate the explanation generation capabilities of Graphon-Explainer through meticulously designed experiments using three real-world datasets and three synthetic datasets. Initially, we train a GNN classifier on each dataset and employ Graphon-Explainer to generate explanations for specific target classes and synthetic motifs that lie close to the decision boundary between two target classes.

## 6.1 Metrics for Quantitative Evaluation of Generated Explanations

Unlike instance-level explanations, model-level explanations lack a corresponding ground truth because it's impossible to determine the exact graph patterns the classifier has learned to distinguish between classes. For generating explanations for individual target classes, the objective is to maximize the class score for the target class when the generated explanation is provided as input to the classifier. For generating graphs that lie on the decision boundary, the objective is to achieve a class score indicating that the generated graphs belong to all classes with equal probability according to the classifier. With this goal in mind, we evaluate our method using the metrics established in the literature(Wang & Shen, 2022; Chen et al., 2024; Yuan et al., 2020) to assess model-level explanations. We report the performance based on the following metrics:

**Target Class Score:** The target class score of a generated explanation represents the probability with which the explanation belongs to the target class, as determined by the classifier $f(.)$. When generating explanations for a target class $c$, each generated explanation is inputted into $f(.)$, and the probability for class $c$ is recorded. We report the mean and standard deviation for 50 generated explanations for each target class. For synthetic graphs generated on the decision boundary, we report the class score of the generated graph across all classes. For target class explanations, a higher class score suggests that the generated explanation includes discriminative motifs identified by the classifier as belonging to the target class, resulting in the classifier confidently classifying the explanation into the target class. Conversely, for synthetic motifs intended to lie close to the decision boundary, the goal is to achieve class scores that assign the generated motif an equal probability of belonging to both the classes.

**Density**: This metric assesses the sparsity of the generated explanations. In a graph $G$, density is calculated as $Density = \frac{|E|}{|V|^2}$, where $E$ represents the edge set and $V$ denotes the vertex set of $G$. It is crucial that explanations generated for a class include only edges that distinctly characterize that class and are essential for classification, thereby minimizing non-vital edges. Lower-density explanations facilitate clearer interpretation for users, enhancing model transparency and trustworthiness. We provide the average density and standard deviation of 50 explanation graphs for each target class.

**Time Efficiency:** Following the evaluation setting in GNNInterpreter (Wang & Shen, 2022), we also note the time taken to generate 10 explanations for each target class and report the mean time taken across all target classes in a particular dataset. We denote this using $T_{10}$ in Table 2.

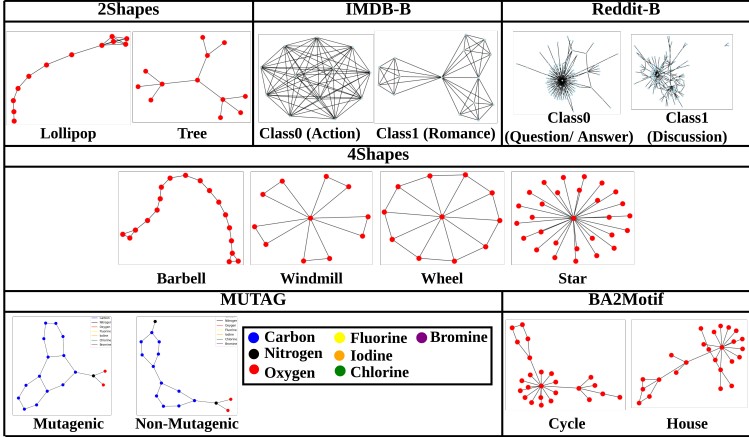

Figure 2: Examples of Graphs in all Target Classes on all Datasets

## 6.2 Metrics for Evaluation of the Decision Boundary of the Classifier

**Boundary Margin:** The boundary margin (Yang et al., 2020) measures the minimum distance between the embeddings of graphs from a class $c_1/c_2$ and those at the boundary between classes $c_1$ and $c_2$. For a class $c_1$, it is defined as:

$$\Psi(f, c_1) = \min_{(G_{c_1}, G_b)} \|\phi(G_{c_1}) - \phi(G_b)\|$$

where $(G_{c_1}, G_b)$ represents a pair of graphs such that $G_{c_1}$ belongs to class $c_1$ by having the highest class score for class $c_1$ according to $f$ and $G_b$ is on the boundary between class $c_1$ and $c_2$. Here, $\phi$ denotes the embedding function of the GNN classifier $f$. A wider boundary margin implies better chance of generalization properties of the classifier since the risk of misclassification decreases during test time. In this paper, for the choice of $\phi$ we use the embedding of the last hidden layer, after which a linear layer is applied to get the class specific logits. This ensures that we compute this measure with linearly separable embeddings.

**Boundary Thickness:** A thicker boundary signifies greater robustness while a thinner boundary is more prone to an adversarial attack. The boundary thickness (Yang et al., 2020) between a target class $c_1/c_2$ and the decision boundary separating the two classes, can be defined for the class $c_1$, as:

$$\Theta(f, \delta, c_1, c_2) = \mathbb{E}_{(G_{c_1}, G_b)} \left[ \|\phi(G_{c_1}) - \phi(G_b)\| \int_0^1 \mathbb{I}\left(\delta > \sigma(h(t))_{c_1} - \sigma(h(t))_{c_2}\right) dt \right]$$

where: $h(t) = (1 - t) \cdot \phi(G_{c_1}) + t \cdot \phi(G_b)$ is the linear interpolation between embeddings, $\sigma(h(t))_{c_1}$ and $\sigma(h(t))_{c_2}$ are the probabilities for classes $c_1$ and $c_2$ respectively, $\mathbb{I}$ is the indicator function, $\delta$ is a threshold, commonly set to 0.75.

**Boundary Complexity:** The boundary complexity(Guan & Loew, 2020) quantifies the intricacy of the data and the classifier's decision-making process by assessing the entropy of the eigenvalue distribution from the covariance matrix of boundary embeddings. It is expressed as:

$$\Xi(g, c_1, c_2) = \frac{H\left(\frac{\mu}{\|\mu\|_1}\right)}{\log K}$$

where: $\mu$ are the eigenvalues of the covariance matrix of the boundary embedding set $\phi(G_b)$, $\|\mu\|_1$ is the L1 norm of $\mu$, $H\left(\frac{\mu}{\|\mu\|_1}\right)$ is the entropy of the normalized eigenvalues, $K$ is the dimensionality of the embeddings. Datasets with real-world complexity or more classes typically exhibit more intricate decision boundaries compared to synthetic datasets or those with fewer classes.

## 6.3 Datasets

We conduct experiments on three synthetic and three real-world datasets.

Table 1: Dataset Properties and Classifier Accuracy

| Dataset | #Classes | #Graphs | Average #Nodes | Average #Edges | Classifier Accuracy |
|---|---|---|---|---|---|
| IMDB-B | 2 | 1000 | 19.77 | 96.53 | 0.7050 |
| Reddit-B | 2 | 2000 | 429.63 | 497.75 | 0.8250 |
| MUTAG | 2 | 188 | 17.93 | 19.79 | 0.88297 |
| 2Shapes | 2 | 200 | 9.32 | 10.37 | 1.0 |
| BA2Motif | 2 | 1000 | 25 | 25.48 | 0.958 |
| 4Shapes | 4 | 4000 | 15.89 | 22.98 | 0.94 |

**Synthetic Datasets:** We generate the **2Shapes** and **4Shapes** datasets. 2Shapes contains graphs belonging to two classes: Lollipop and Tree. 4Shapes contains graphs belonging to 4 classes: Windmill, Wheel,

Star, and Barbell. A graph belonging to a particular class has the shape as described by the name of the corresponding class. Examples of graphs in all the target classes are shown in Figure 2. The method to generate the datasets is detailed in Appendix A. We also demonstrate the effectiveness of our method on the **BA2Motif** (Luo et al., 2020) dataset which consists of graphs with two classes of motifs: house and cycle. **Real Datasets:** For real datasets, we conduct experiments using the IMDB-BINARY (**IMDB-B**), REDDIT-BINARY (**REDDIT-B**), and **MUTAG** datasets. The **IMDB-B** dataset comprises 1,000 movie collaboration networks, where nodes represent actors or actresses and edges denote co-appearances in movies. The graphs are categorized into Action or Romance genres. The **REDDIT-B** dataset represents Reddit discussions, where nodes correspond to users and edges indicate replies. Each graph is labeled as belonging to either a question/answer-based or discussion-based community. The **MUTAG** dataset contains chemical compound graphs, with nodes as atoms and edges as bonds, labeled as either mutagenic or non-mutagenic. Examples of graphs from all datasets are shown in Figure 2. Dataset properties and classifier test accuracy are summarized in Table 1, with the architecture of the GNN classifiers detailed in Appendix B.

## 6.4 Results

We assess the performance of Graphon-Explainer both qualitatively and quantitatively. We also demonstrate comparative results by comparing our method with GNNInterpreter and XGNN. Our code repository is provided at `https://github.com/amisayan/Graphon-Explainer`. Results on all metrics are presented in Table 2 for synthetic datasets and real datasets. An overarching trend observed across almost all datasets is that Graphon-Explainer can generate explanations around 10 times faster than GNNInterpreter. Table 3 presents the results for motifs generated near the decision boundary. The class scores clearly shows that Graphon-Explainer produces boundary motifs with competitive, or even superior, boundary scores compared to those generated by GNNBoundary. It should be noted that the decision boundary metrics in Table 3 with respect to the classifier are computed using the motifs generated by Graphon-Explainer. Qualitative results in Figure 3a and 3b show the estimated graphon for each target class and the corresponding explanation generated by Graphon-Explainer and GNNInterpreter. Since XGNN, on most classes generates a single node or a line graph as explanations, the generated explanations are deferred to Figure 6 in the Appendix F. The boundary motifs generated by GNNBoundary is also deferred to the Appendix K. It can be observed that the estimated graphons of each target class clearly look different. Hence, each class has a distinct motif based on which the classifier categorizes them into separate classes. The visualization of the estimated graphons of different target classes in Figure 3a and 3b gives added insight into the structure of graphs contained in various target classes and help make the generative model itself more interpretable.

Table 2: Quantitative Results

| | Dataset | Classes | Graphon-Explainer | | | GNNInterpreter | | | XGNN | | |
|---|---|---|---|---|---|---|---|---|---|---|---|
| | | | Class Score | Density | $T_{10}$ (seconds) | Class Score | Density | $T_{10}$ (seconds) | Class Score | Density | $T_{10}$ (seconds) |
| Synthetic Dataset | 2Shapes | Lollipop | **1.0 ± 0.00** | 0.1481 ± 0.021 | **0.34** | 0.8576 ± 0.271 | **0.088 ± 0.041** | 17.23 | 0.999 ± 0.003 | 0.25 ± 0.00 | 415.918 |
| | | Tree | **1.0 ± 0.00** | **0.1389 ± 0.034** | | 0.9123 ± 0.042 | 0.2353 ± 0.0140 | | 0.0010 ± 0.00 | 0.0 ± 0.00 | |
| | BA2Motif | House | **1.0 ± 0.00** | **0.0667 ± 0.0042** | **0.47** | 0.9812 ± 0.0002 | 0.1823 ± 0.0234 | 22.35 | 0.4918 ±0.0001 | 0.25 ± 0.00 | 396.96 |
| | | Cycle | **1.0 ± 0.00** | **0.0663 ± 0.0034** | | 0.9475 ± 0.0006 | 0.3821 ± 0.0416 | | 0.5027 ±0.00 | 0.25 ± 0.00 | |
| | 4Shapes | Barbell | **0.96 ± 0.0001** | **0.0902 ± 0.0036** | **0.72** | 0.5921 ± 0.0023 | 0.1423 ± 0.1033 | 15.78 | 1.0 ±0.00 | 0.25 ± 0.00 | 328.05 |
| | | Windmill | **1.0 ± 0.00** | **0.1067 ± 0.0240** | | 0.6713 ± 0.2423 | 0.1260 ± 0.0744 | | 0.0 ± 0.0001 | 0.0 ± 0.00 | |
| | | Wheel | **0.98 ± 0.0002** | 0.1111 ± 0.0315 | | 0.8521 ± 0.0102 | **0.1100 ± 0.0245** | | 0.0 ± 0.0007 | 0.25 ± 0.00 | |
| | | Star | **1.0 ± 0.00** | **0.0663 ± 0.0038** | | 0.99 ± 0.0001 | 0.1705 ± 0.0303 | | 0.0 ± 0.0000 | 0.0 ± 0.00 | |
| Real Dataset | Mutag | Mutagenic | **1.0 ± 0.00** | **0.0675 ± 0.0047** | **2.65** | 0.99 ± 0.0031 | 0.1044 ± 0.0565 | 2.75 | **1.0 ± 0.00** | 0.1523 ± 0.0039 | 198.61 |
| | | Non-Mutagenic | **1.0 ± 0.00** | **0.0900 ± 0.0032** | | 0.9691 ± 0.0012 | 0.1566 ± 0.0687 | | **1.0 ± 0.00** | 0.1536 ± 0.0012 | |
| | IMDB-Binary | Class0 (Action) | **1.0 ± 0.00** | 0.2158 ± 0.0191 | **1.34** | 0.6500 ± 0.0201 | 0.2461 ± 0.0135 | 254.32 | 0.4990 ± 0.00 | 0.0 ± 0.00 | 409.57 |
| | | Class1 (Romance) | **1.0 ± 0.00** | **0.2094 ± 0.0314** | | 0.3540 ± 0.0201 | 0.2193 ± 0.061 | | 0.5320 ± 0.00 | 0.0 ± 0.00 | |
| | Reddit-Binary | Class0 (Question/ Answer) | **0.9834 ± 0.0002** | 0.0168 ± 0.0007 | **2.48** | 0.8216 ± 0.0142 | **0.0154 ± 0.0021** | 45.15 | 0.004 ± 0.00 | 0.3056 ± 0.0482 | 356.78 |
| | | Class1 (Discussion) | 0.9827 ± 0.0001 | **0.0170 ± 0.0018** | | **0.9889 ± 0.0000** | 0.0173 ± 0.0003 | | 0.99 ± 0.00 | 0.0 ± 0.00 | |

**Synthetic Datasets:** The quantitative results in Table 2 demonstrate that Graphon-Explainer consistently outperforms GNNInterpreter and XGNN across nearly all metrics. This indicates that Graphon-Explainer is

Table 3: Quantitative Results on all Datasets for the Decision Boundary

| Dataset | $c_1$ | $c_2$ | Boundary Metrics of the Classifier | | | | | Graphon-Explainer | | GNNBoundary | |
|---|---|---|---|---|---|---|---|---|---|---|---|
| | | | Boundary Margin | | Boundary Thickness | | Boundary Complexity | Class Scores | | Class Scores | |
| | | | $c_1$ | $c_2$ | $c_1$ | $c_2$ | | $p(c_1)$ | $p(c_2)$ | $p(c_1)$ | $p(c_2)$ |
| **2Shapes** | Lollipop | Tree | 1.6910 | 1.7109 | 0.6864 | 2.733 | 0.00844 | **0.4912 ± 0.0002** | **0.5123 ± 0.0004** | 0.2988 ± 0.3502 | 0.7010 ± 0.3502 |
| **BA2Motif** | House | Cycle | 0.214 | 0.300 | 0.9988 | 1.91867 | 0.1380 | 0.5315 ± 0.0480 | **0.4836 ± 0.0402** | **0.4691± 0.0841** | 0.5285 ± 0.0890 |
| **IMDB-B** | Class0 (Action) | Class1 (Romance) | 1.0035 | 0.8864 | 1.6602 | 3.7237 | 0.2246 | 0.5136 ± 0.0002 | 0.5023 ± 0.0001 | **0.4994 ± 0.0000** | **0.5005 ± 0.0000** |
| **Reddit-B** | Class0 (Question/ Answer) | Class1 (Discussion) | 0.2739 | 0.31 | 1.84 | 2.44 | 0.132 | **0.5005 ± 0.0000** | **0.5003 ± 0.0000** | 0.4883 ± 0.0227 | 0.5184 ± 0.0315 |
| **Mutag** | Mutagenic | Non-mutagenic | 29.303 | 30.746 | 43.896 | 34.811 | 0.096 | **0.5014 ± 0.0522** | **0.4982 ± 0.0418** | 0.2896 ± 0.3962 | 0.7103 ± 0.3964 |
| **4Shapes** | Barbell | Wheel | 0.7243 | 0.9656 | 5.64008 | 3.6377 | 0.02906 | **0.4762 ± 0.0302** | **0.4812 ± 0.0291** | 0.4333 ± 0.1465 | 0.5584 ± 0.1563 |
| | Barbell | Star | 0.8292 | 2.807 | 7.3156 | 0.5376 | 0.07444 | **0.5001 ± 0.0989** | **0.4124 ± 0.0992** | 0.2965 ± 0.3129 | 0.5781 ± 0.2941 |
| | Windmill | Wheel | 0.4993 | 0.4907 | 2.4315 | 4.6429 | 0.0855 | **0.4995 ± 0.0010** | **0.5002 ± 0.0014** | 0.7953 ± 0.1527 | 0.1695 ± 0.1485 |
| | Windmill | Star | 0.4511 | 3.803 | 6.140 | 0.596 | 0.01686 | **0.4812 ± 0.0232** | **0.4882 ± 0.0196** | 0.8876 ± 0.0021 | 0.1122 ± 0.0460 |

not only capable of generating discriminative motifs that closely align with the classifier's learned patterns but does so more efficiently, producing sparser explanations in significantly less time than existing methods.

The qualitative analysis in Figure 3a further supports these findings, showing that the explanations generated by Graphon-Explainer align well with the class identities when the classifier performs accurately, as seen with the 2Shapes dataset. The classification task on this dataset is relatively straightforward for the classifier, as evidenced by the minimal boundary complexity shown in Table 3 and the confusion matrix in Figure 4.

However, the 4Shapes dataset presents a more challenging scenario. While the explanations for the Barbell, Windmill, and Wheel classes may not perfectly match the class identities, they still achieve high class scores, suggesting a greater likelihood of misclassification for these classes. Analysis of the adjacency scores in Figure 5 and decision boundary metrics in Table 3 reveals that within the 4Shapes dataset, the Windmill and Wheel classes share the thinnest boundary margins and the highest boundary complexity. Additionally, the Windmill class has a thinner boundary in its decision boundary with the Wheel class, indicating a higher propensity for misclassification as Wheel rather than the other way around. This suggests that the embeddings of these classes are more closely aligned, making it more difficult for the classifier to distinguish between them, as reflected in both the confusion matrix in Figure 4 and the similarity of their explanations in Figure 3a.

The highest boundary complexity among the synthetic datasets is observed for the BA2Motif dataset, likely due to the more complex task of classifying random graphs based on the presence of a specific motif—a problem that is inherently more challenging due to the noise within these graphs. Moreover, graphs containing a House motif are more likely to be misclassified as Cycle, as indicated by the thinner boundary margin for the House class. This finding is corroborated by the confusion matrix in Figure 4. Finally, an analysis of the motifs generated near the decision boundary in Figure 3a shows that motifs close to the boundary between two target classes often represent a mixture of motifs from both classes.

**Real Datasets:** The results in Table 2 clearly highlight the superiority of Graphon-Explainer over existing methods. A qualitative analysis of the explanations from Figure 3b on the MUTAG dataset shows that Graphon-Explainer can generate more realistic molecular features compared to GNNInterpreter. It's worth noting that XGNN also performs well on this dataset, producing more realistic explanations than GNNInterpreter, as illustrated in Figure 6. For the Mutagenic class, Graphon-Explainer's explanations include the fused ring structure and the $NO_2$ subgroup—key distinguishing features of this class—whereas these are absent in other explanations. For the Non-Mutagenic class, no **O** atom is present in the explanations, aligning with the class identity. An analysis of the boundary metrics from Table 3 reveals that the Non-Mutagenic class has a lesser boundary thickness, which increases the likelihood of misclassification of Non-Mutagenic compounds as Mutagenic. This observation is supported by the confusion matrix in Figure 4. Moreover, the boundary complexity score is the lowest on this dataset compared to other real datasets, indicating that the decision rules for class assignment on the MUTAG dataset are simpler.

On the **REDDIT-B** dataset, both Graphon-Explainer and GNNInterpreter perform well, as shown in Table 2. Explanations for Class 0 (the Question-Answer class) in Table 3b reveal a pattern of one high-degree

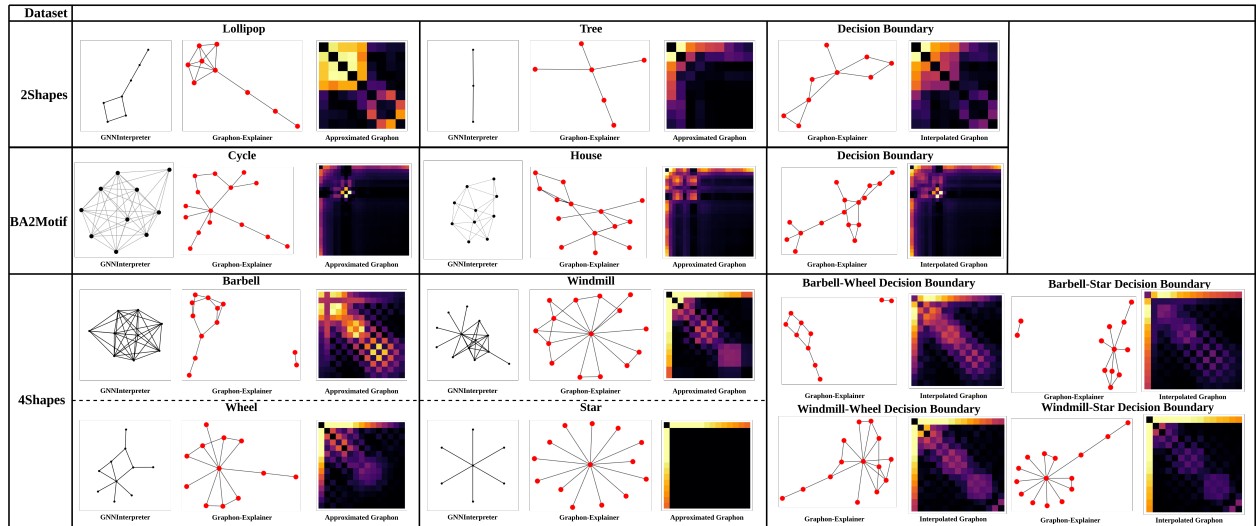

(a) Synthetic Dataset

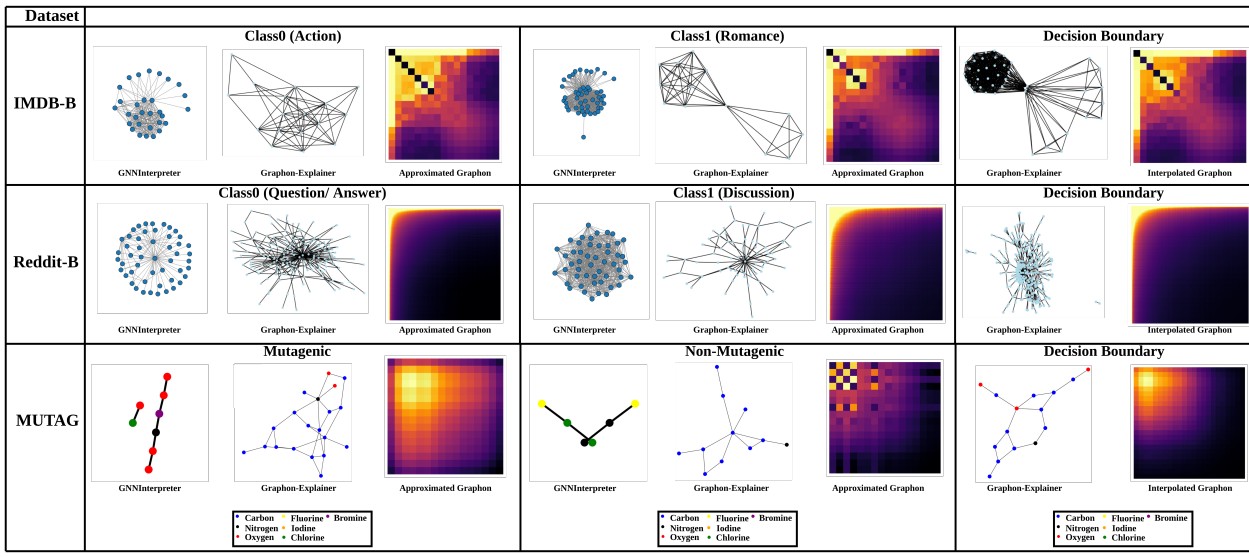

(b) Real Dataset

Figure 3: Qualitative results across all datasets. Graphon-Explainer's explanations have class scores of 1, and boundary motifs score close to 0.5. The top-scoring GNNInterpreter explanations are also shown. In the approximated graphons, colors transition from dark to bright, reflecting values from 0 to 1, with brighter colors indicating higher values.

node connected to several low-degree nodes, capturing the characteristic of a single user answering multiple questions—a key feature of this class. In contrast, explanations for Class 1 (the Discussion class) show a few users frequently interacting among themselves, with other users on the fringe engaging with only a single user, typical of discussion groups. Boundary analysis from Table 3 indicates that the boundary margin and thickness are both smaller for the Question-Answer class, increasing the risk of misclassification as belonging to the Discussion class. This finding is corroborated by the classifier's confusion matrix in Figure 4.

On the **IMDB-B** dataset, Graphon-Explainer comprehensively outperforms competing methods, as demonstrated in Table 2. Explanations for Class 0 (Action genre) in Table 3b feature the signature motif of multiple co-actors who have previously worked together, reflecting the typical structure of an Action movie with multiple popular actors in key roles. In contrast, explanations for Class 1 (Romance genre) show a distinct motif

of one popular actor interacting with multiple groups of actors, consistent with the genre's focus on one or two central characters. The boundary graph appears to combine these two distinct motifs. Additionally, boundary metrics analysis indicates that this task has the highest boundary complexity, suggesting that the classifier's decision-making process to distinguish between these two classes of graphs is particularly complex. Furthermore, graphs in the Action class are more prone to misclassification due to lesser boundary thickness, which is validated by the confusion matrix in Figure 4.

## 6.5 Sensitivity Analysis

Figure 12 in Appendix L illustrates how class scores vary with changes in Graphon-Explainer's hyperparameters as listed in Algorithm 1. For explanations of individual target classes, we adjust the number of partitions $K$ used to approximate the graphon, which alters the number of nodes in the generated graphs. For graphs generated to belong close to the decision boundary, we vary the linear interpolation parameter $\lambda$ and report the class score variations for a single class. We also detail the $n_{sample}$ hyperparameter used for each dataset in Table 8 in Appendix L. It can be concluded from the plots in Figure 12 and Table 8 that the target class scores for explanations do not vary much with change in $K$, class scores for boundary motifs remain in the desirable range when $\lambda$ is around 0.5 and $n_{samples}$ used is reasonably low which makes our method robust to hyperparameter changes.

## 7 Conclusion

This study introduces Graphon-Explainer, a model-level explanation approach for GNNs that leverages graphons, approximating one for each target class, to serve as generators for discriminative motifs identified by the GNN as indicative of each class. Unlike existing model-level techniques limited to generating explanations for individual target classes, Graphon-Explainer transcends this constraint by producing synthetic motifs close to the decision boundary of two target classes. This capability not only elucidates what the GNN has learned for each class but also aids in delineating the decision boundary of the trained GNN. Through extensive experiments and theoretical analysis, we demonstrate that Graphon-Explainer faithfully elucidates any GNN, highlighting its strengths and pitfalls in graph classification tasks. Notably, it achieves this while generating diverse explanations significantly faster than current state-of-the-art methods.

We believe Graphon-Explainer can be used in many real-world applications to inform users about the inductive biases and underlying classification logic of a GNN model deployed in practical scenarios. For example, our experiment on the MUTAG dataset demonstrates how the classifier identifies fused rings as key motifs for mutagenic classes. Similarly, our method could explain a GNN classifying toxic and non-toxic drugs, revealing the basis on which the GNN classifies a molecule as toxic. Analyzing the decision boundary of such a GNN could provide critical insights into the model's robustness and the risk of misclassifying toxic molecules as non-toxic.

Furthermore, as demonstrated by additional experiments in Appendix H, even when two classifiers perform accurately, their interpretations of a single class can vary significantly based on their inductive biases. This insight can help users choose between classifiers with equal accuracy based on their unique interpretative properties in real-world settings. Moreover, our approach can generate explanations tailored to the specific dataset a user possesses since estimated graphons effectively capture the dataset's distribution. This personalized approach offers tailored insights depending on the types of instances present in a user's query dataset, rather than providing uniform explanations regardless of dataset variations.

Overall, Graphon-Explainer not only advances our understanding of GNN behavior but also equips users with a powerful tool to assess, interpret, and trust the decisions made by GNNs in diverse application domains.

### Acknowledgments

SB acknowledges the JC Bose Fellowship grant No. JBR/2021/000036/SSC from ANRF-SERB, Govt. of India.

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

## A    Synthetic Dataset Generation

We detail the method to generate the datasets 2Shapes and 4Shapes in Algorithm 2 and Algorithm 3 respectively.

---

**Algorithm 2** 2shapes dataset generation

---

**Graph Classes:** {LOLLIPOP, TREE}
**Output:** A collection of pairs consisting of a graph and its label
**Procedure:**
- For every class, $C \in$ Graph Classes, do 1000 times:
    - if $C$ = LOLLIPOP:
        * Sample lollipop graph $G_{lollipop}$ with random number of head nodes $m \in \{2, \cdots, 6\}$ and random number of tail nodes $k \in \{4, \cdots, 16\}$
        * Return $(G_C, C)$.
    - if $C$ = TREE:
        * Sample a r-ary tree $G_{tree}$ with random $r \in \{2, 3, 4\}$ and random number of nodes $n \in \{3, 4, \cdots, 13\}$
        * Return $(G_C, C)$.

---

**Algorithm 3** 4Shapes dataset generation

---

**Graph Classes:** {BARBELL, WINDMILL, WHEEL, STAR}
**Output:** A collection of pairs consisting of a graph and its label
**Procedure:**
- For every class, $C \in$ Graph Classes, do 1000 times:
    - if $C$ =BARBELL:
        * Sample barbell graph $G_{Barbell}$ with random number of head nodes $m, n \in \{1, \cdots, 4\}$ and random number of connector nodes $k \in \{4, \cdots, 15\}$
        * Return $(G_C, C)$.
    - if $C$ = WINDMILL:
        * Sample windmill graph $G_{windmill}$ with random number of cliques $k \in \{3, \cdots, 9\}$ and each clique of random size $s \in \{3, \cdots, 5\}$
        * Return $(G_C, C)$.
    - if $C$ = WHEEL:
        * Sample wheel graph $G_{wheel}$ with a central node and random number of non-central nodes $n \in \{4, \cdots, 16\}$
        * Return $(G_C, C)$.
    - if $C$ = STAR:
        * Sample star graph $G_{star}$ with random number of non-center nodes $n \in \{6, \cdots, 30\}$
        * Return $(G_C, C)$.

---

## B   Classifier Architecture

On all datasets except MUTAG, we implement a deep GCN architecture comprising three GCN layers, each with a dimension of 64. This is followed by a global mean pooling layer, a batch normalization layer, and finally a linear layer to obtain the class logits. We utilize Leaky ReLU as the activation function between the GCN layers and within the MLP. The model parameters are initialized using the Kaiming method. Training is conducted using the Adam optimizer (Kingma & Ba, 2014) with an initial learning rate of 0.01. Additionally, we use a learning rate scheduler to halve the learning rate every hundred epochs. On the MUTAG dataset, the model architecture consists of three graph convolutional layers designed to extract features from input graphs. The first and third layers are Graph Convolutional Networks (GCN), while the second layer is a Graph Attention Network (GAT) with two attention heads, enabling the model to focus on important nodes and edges. A residual connection from the input features to the output of the third layer ensures better gradient flow and feature retention. The model employs a Jumping Knowledge mechanism in concatenation mode, aggregating features from all three layers to capture hierarchical information. The aggregated features are globally pooled, batch normalized, and passed through a dropout layer with a 0.5 probability to prevent overfitting. Finally, these processed features are fed into a linear classifier that outputs the class logits. This architecture is trained the same way as the above architectures.

The confusion matrix for each of the trained classifier is shown in Figure 4.

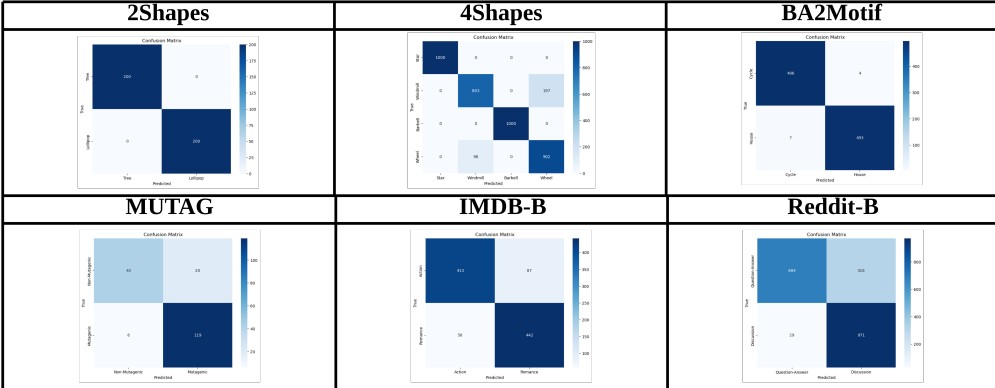

Figure 4: Confusion Matrix of the Trained Classifier on all Datasets

## C   Graphon Estimation and Related Guarantees

The algorithm for estimating the graphon of a target class is detailed in Algorithm **??**.

The weak regularity lemma for graphons(Lovász, 2012) can be stated as

**Theorem C.1.** *Let $S$ be any graphon and let $K \geq 1$, $K \in \mathbb{Z}$. Then, there exists a step function $S_P$ over a partition $P = \{P_1, \cdots, P_K\}$ of the unit interval such that $\|S - S_P\|_\square \leq \frac{2}{\sqrt{\log K}} \|S_P\|_2$*

This theorem guarantees that there would always exist a step function and a partition with which any given graphon can be approximated well in the cut norm. Also note that by increasing the number of partitions, one can arbitrarily increase the accuracy of the approximation.

## D   Theoretical Guarantees

### D.1   Proofs of the Theorems

**Theorem D.1.** *Let $D_{c_1}$ and $D_{c_2}$ be two collections of graphs belonging to the target classes $c_1$ and $c_2$, respectively, with their estimated graphons $S^{c_1}$ and $S^{c_2}$. Define the boundary graphon as $S^b = \lambda S^{c_1} + (1 - $*

---

**Algorithm 4** Estimating the Graphon of a Target Class and Generating Graphs from the Graphon

---

**Input:** Set of graphs $D_c$ belonging to class $c$, threshold $\tau$ for clipping singular values, desired number of nodes $N$ (where $N$ must not exceed the maximum number of nodes in any graph from $D_c$)
**Init:** List of sorted adjacency matrices $\mathbb{A}$, List of sorted node features $\mathbb{X}$
**Output:** Estimated graphon $S^c$ for class $c$, Node feature of the generated graph $X_c$, Adjacency matrix of the generated graph $A_c$
**Procedure:**
- **Align and Sort Nodes:**
  - For each graph $G = (A, X)$ in the set $D_c$:
    * Compute node degrees from the adjacency matrix $A$.
    * Sort nodes in descending order of their degrees to obtain the sorted adjacency matrix $\overline{A}$ and sorted node features $\overline{X}$.
    * Pad $\overline{A}$ to size $N \times N$ and $\overline{X}$ to size $N \times d$, where $d$ is the node feature dimension.
    * Append the sorted and padded adjacency matrix $\overline{A}$ to the list $\mathbb{A}$.
    * Append the sorted and padded node features $\overline{X}$ to the list $\mathbb{X}$.
- **Graphon Estimation using USVT:**
  - Compute the mean adjacency matrix sum\_graph $= \frac{1}{|\mathbb{A}|} \sum_{i=1}^{|\mathbb{A}|} \overline{A}_i$, where $\overline{A}_i$ represents each sorted adjacency matrix in the set $\mathbb{A}$.
  - Perform Singular Value Decomposition (SVD) on sum\_graph: sum\_graph $= U\Sigma V^T$.
  - Apply singular value thresholding: set $\sigma_i = 0$ if $\sigma_i < \tau \times \sqrt{N}$.
  - Reconstruct the graphon $S^c = U\Sigma V^T$ using the modified singular values.
  - Clip the values of $S^c$ to the range $[0, 1]$.
- **Sampling Graphs from the Estimated Graphon:**
  - For each generation of a graph using the graphon, repeat the following steps to generate its node feature $X_c$:
    * For $i = \{1, \cdots, N\}$:
      · Collect the $i^{th}$ row from all matrices $\overline{X}$ in $\mathbb{X}$ into a pool.
      · Assign a probabilistic weight to each unique row in the pool based on its frequency of occurrence, excluding rows that are all zeroes to avoid padding effects.
      · Randomly sample the $i^{th}$ row of the graph's node feature using the assigned probabilistic weights.
    * Combine the sampled rows to form the node feature $X_c$ of the generated graph.
  - Sample $S_c$ using Equation 4 to generate the adjacency matrix $A_c$ of the sampled graph.
**Return:** $S^c$: Estimated graphon for class $c$, $X_c$: Node feature of the sampled graph, $A_c$: Adjacency Matrix of the sampled graph

---

$\lambda)S^{c_2}$. *For any discriminative motif $M_{c_1} \in \mathbb{M}_{c_1}$ and $M_{c_2} \in \mathbb{M}_{c_2}$, the difference in the homomorphism density of $M_{c_1}$ and $M_{c_2}$ with the boundary graphon $S^b$ compared to their respective graphons $S^{c_1}$ and $S^{c_2}$ is bounded above by:*

$$|t(M_{c_1}, S^{c_1}) - t(M_{c_1}, S^b)| \leq (1 - \lambda)|E(M_{c_1})|\|S^{c_1} - S^{c_2}\|_\square$$
$$|t(M_{c_2}, S^{c_2}) - t(M_{c_2}, S^b)| \leq \lambda|E(M_{c_2})|\|S^{c_1} - S^{c_2}\|_\square$$

*Proof.* We use the following lemma from (Lovász & Szegedy, 2006) for the proof of this theorem.

**Lemma D.2.** *Let $U$, $W$ be two graphons. Then, for every simple graph $F$ $|t(F, U) - t(F, W)| \leq |E(F)|\|U - W\|_\square$*

Using D.2 we see that, $|t(M_{c_1}, S^{c_1}) - t(M_{c_1}, S^b) \leq |E(M_{c_1})|\|S^{c_1} - S^b\|_\square$
This can be rewritten as :

$$|E(M_{c_1})|\|S^{c_1} - S^b\|_\square = |E(M_{c_1})|\|S^{c_1} - \lambda S^{c_1} - (1 - \lambda)S^{c_2}\|_\square$$
$$= |E(M_{c_1})|\|(1 - \lambda)(S^{c_1} - S^{c_2})\|_\square$$

Now, note that for any graphon $S$ and a positive constant $\alpha \in \mathbb{R}$, the cut norm has the following property $\|\alpha S\|_\square = \sup_{X,Y \subset [0,1]} \left|\int_{X \times Y} \alpha S(x, y)\right| = \alpha \sup_{X,Y \subset [0,1]} \left|\int_{X \times Y} S(x, y)\right| = \alpha\|S\|_\square$
Using this property, we conclude,

$$|E(M_{c_1})|\|(1 - \lambda)(S^{c_1} - S^{c_2})\|_\square = (1 - \lambda)|E(M_{c_1})|\|S^{c_1} - S^{c_2}\|_\square$$

The proof for a motif $M_{c_2}$ can be done similarly by mimicking this proof $\qquad\square$

Next, we present a theoretical guarantee that graphs generated from the graphon $S^b$ will inherit the motifs contained in $S^b$.

**Theorem D.3.** *Let $S^b$ be the boundary graphon and assume that it contains a discriminative motif $M_b$. Then any random graph $G$ on $n$ nodes generated from $S^b$ satisfies:*

$$P(|t(M_b, G) - t(F, S^b)| > \epsilon) \leq 2 \exp\left(-\frac{\epsilon n^2}{18|V(M_b)|^2}\right)$$

*Proof.* We begin by stating a lemma from Lovász (2006) (Lovász & Szegedy, 2006), from which the proof of our theorem will follow as a straightforward corollary.

**Lemma D.4.** *Let $S$ be a graphon and $F$ be a simple graph. Let $G(n, S)$ be a random graph on $n$ nodes sampled from the graphon $S$. Then, $P(|t(F, G) - t(F, S)| > \epsilon) \leq 2 \exp\left(-\frac{\epsilon n^2}{18|V(F)|^2}\right)$*

Setting, $F = M_b$ and $S = S^b$ in Lemma D.4, the bound follows $\qquad\square$

Theorem 5.2 demonstrates that the topology of a random graph sampled from the boundary graphon closely resembles the topology of the boundary graphon itself, which includes discriminative motifs from both classes. Consequently, the theoretical analysis in this section confirms that graphs generated by the boundary graphon $S^b$ will contain discriminative motifs from both target classes.

### D.2 Generation of Diverse Explanations

The estimated graphon $S^c$ of a target class $c$ is used to generate explanations for the target class $c$. When the graphon is approximated using a step function over $K$ partitions of the unit interval, then $S^c \in \mathbb{R}^{K \times K}$, from which the adjacency matrix $A$ is sampled as $A_{ij} \sim \text{Bern} S_{ij}^c$. Consequently, the probability of generating two identical graphs using this sampling scheme is $\prod_{i=1}^{K} \prod_{j=1}^{K} ((S_{ij}^c)^2 + (1 - S_{ij}^c)^2)$ which is extremely small since $S_{ij}^c \in [0, 1]$ and $K$ is a reasonably large number $K$ is set to be the median number of nodes of graphs contained in the dataset $D$. This guarantees that Graphon-Explainer produces explanations that are distinct from each other.

### D.3 Time Complexity Analysis

The estimation of a graphon using the USVT method on $g$ graphs, each with $N$ nodes, and employing a step function with $K$ partitions, has a time complexity of $O(N^3)$ (Xu et al., 2021). After optimization, generating a single explanation from $s$ samples of the estimated graphon requires $O(sK)$ time for node feature generation and $O(sK^2)$ for edge generation, resulting in a total explanation generation complexity of $O(sK^2)$.

This is more efficient than GNNInterpreter, which has a time complexity of $O(TsK^2)$ for a single explanation, where $T$ is the number of iterations needed for convergence, and $s$ represents the number of Monte Carlo samples. Notably, our method is non-iterative and does not require convergence for each explanation, as the graphon is estimated once for a target class and then sampled from, making our approach significantly faster than GNNInterpreter.

Moreover, the XGNN method exhibits a considerably higher time complexity of $O(TRM^3)$, where $T$ is the number of episodes, $R$ is the number of rollout operations per episode step, and $M$ is the number of edges in the generated graph. Given that $M$ typically exceeds $K - 1$, XGNN's time complexity is substantially worse.

## E Adjacency Score in a Multi-Class Scenario

### E.1 Algorithm for Computing Adjacency Score

---

**Algorithm 5** Adjacency Score Computation

---

**Input:**
- $L$: Last linear layer of the classifier that outputs class specific logits
- Embeddings $\mathcal{E}_{c_1}$: Set of embeddings for class $c_1$.
- Embeddings $\mathcal{E}_{c_2}$: Set of embeddings for class $c_2$.
- Class indices $c_1$, $c_2$: Indices of the two target classes for adjacency check.
- Number of samples $N$: Number of samples to draw from each list for checking.
- Number of interpolation steps $T$: Number of steps for linear interpolation.

**Procedure:**
- Initialize adjacency score adjscore $\leftarrow 0$
- **for** $i = 1$ to $N$:
    - Initialize score $S \leftarrow 1$
    - Randomly sample $e_{c_1} \in \mathcal{E}_{c_1}$ and $e_{c_2} \in \mathcal{E}_{c_2}$
    - **for** $\alpha = 0$ to 1 with step size $\frac{1}{T}$:
        - * Compute linear combination $\mathbf{c} \leftarrow \alpha \cdot e_{c_1} + (1 - \alpha) \cdot e_{c_2}$
        - * Pass $\mathbf{c}$ through the model: $\mathbf{p} \leftarrow \text{softmax}(L(\mathbf{c}))$
        - * **if** $\text{argmax}(\mathbf{p}) \notin \{c_1, c_2\}$:
            - · Set $S \leftarrow 0$
            - · **break**
- adjscore = adjscore + $S$
- **return** adjscore/N

---

In this section, we detail in Algorithm 5 the computation of adjacency scores for two target classes in a multi-class scenario. If the score is above a certain threshold, we consider two classes to be adjacent. We use a threshold of 0.75 in this paper.

## E.2 Adjacency Score on 4Shapes

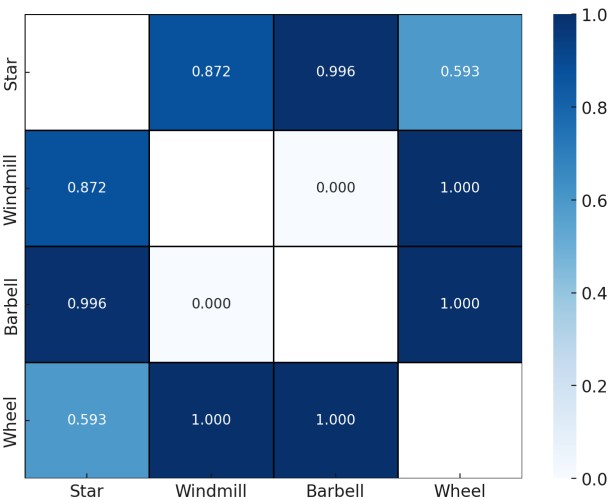

Figure 5: Adjacency Scores between all classes in 4Shapes dataset

The confusion matrix depicting the scores which is used for choosing the boundary in 4Shapes dataset is shown in Figure 5.

## F XGNN Results

The visual results for XGNN is shown in Figure 6.

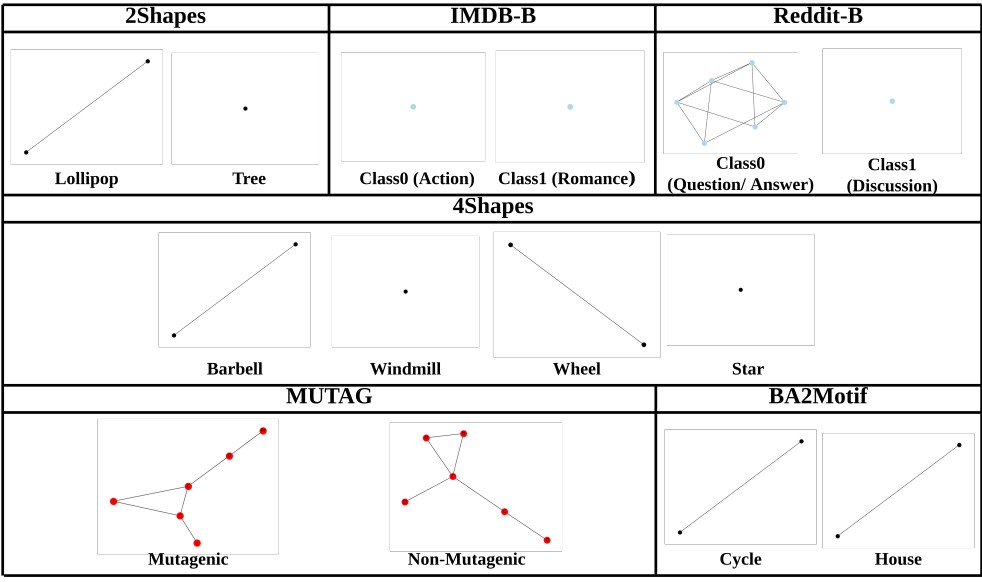

Figure 6: Visual Results from XGNN for real and synthetic dataset

## G   Code Usage for Comparison

We use the code for XGNN from the DIG repository `https://github.com/divelab/DIG/tree/main/dig/xgraph/XGNN`. We use the code for GNNInterpreter from their official repository `https://github.com/yolandalalala/GNNInterpreter` and the code from this reproducibility study(Vasilcoiu et al., 2024) `https://github.com/MeneerTS/FACT2024_GNNInterpreter`.

## H   Analysis of Explanations Generated by Different Architectures

Table 4: Boundary Metrics for the GIN architecture

| Boundary Margin | | Boundary Thickness | | Boundary Complexity |
|---|---|---|---|---|
| Windmill | Wheel | Windmill | Wheel | |
| 4.1341 | 4.7986 | 3.1721 | 9.2698 | 0.21000 |
| Barbell | Wheel | Barbell | Wheel | |
| 3.2143 | 2.9876 | 5.6178 | 5.2197 | 0.14100 |
| Windmill | Barbell | Windmill | Barbell | |
| 1.2149 | 0.6987 | 1.4561 | 3.8654 | 0.09870 |
| Star | Barbell | Star | Barbell | |
| 0.4129 | 3.9851 | 5.7614 | 3.2198 | 0.06842 |
| Windmill | Star | Windmill | Star | |
| 2.1965 | 7.2684 | 2.4976 | 8.6514 | 0.10820 |

Table 5: Boundary metrics for the GCN architecture

| Boundary Margin | | Boundary Thickness | | Boundary Complexity |
|---|---|---|---|---|
| Barbell | Wheel | Barbell | Wheel | |
| 0.7243 | 0.9656 | 5.64008 | 3.6377 | 0.02906 |
| Barbell | Star | Barbell | Star | |
| 0.8292 | 2.807 | 7.3156 | 0.5376 | 0.07444 |
| Windmill | Wheel | Windmill | Wheel | |
| 0.4993 | 0.4907 | 2.4315 | 4.6429 | 0.0855 |
| Windmill | Star | Windmill | Star | |
| 0.4511 | 3.803 | 6.14 | 0.596 | 0.01686 |

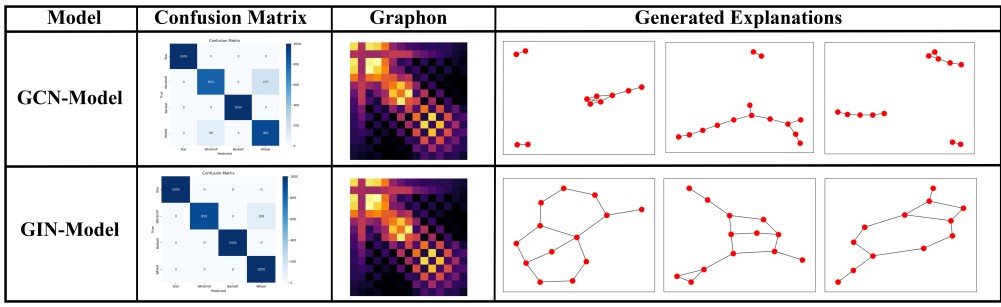

Figure 7: Explanations on the Barbell Class of the 4Shapes dataset for 2 different classifier architectures.

In this section, we demonstrate how different two classifiers are in their intrinsic interpretation of a target class and their decision boundaries even in the scenario when they classify a dataset almost perfectly and even in the case when they have a 100% accuracy on some classes in the dataset. For this, we use the previous GCN architecture detailed in Appendix B and a GIN of the same architecture on the 4Shapes dataset.

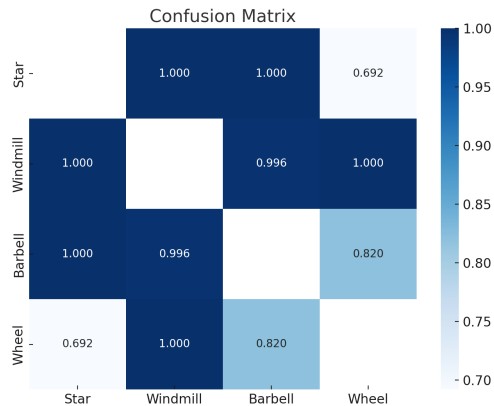

Figure 8: Adjacency Scores on 4Shapes for the GIN model

As seen in Figure7, even though both the classifier have a 100% accuracy on the Barbell class and the learned graphon generative model is the same, sampling using equation 2 yields very different explanations. We find that the GCN model identifies Barbell using the branch like structures of the Barbell while the GIN model distinguishes the barbell class using the head like structure of the barbell. Also, as shown in Figure8 and Figure 5 different pairs of classes share decision boundaries for both models as their inherent graph embeddings are different. Further, even when a pair of common classes share a decision boundary, the boundary metrics are very different( demonstrated in Table 4 and 5) like in the case of Star and Barbell where the boundary thickness of Star is greater for the GIN and the thickness for Barbell is greater for the GCN architecture. However, they also share certain common grounds like the boundary complexity for the Windmill-Wheel pair is the highest for both the models indicating that separating these two classes is significantly harder. This is also validated by the confusion matrices of both the models in Figure 7.

## I  Smaller Explanations for IMDB-B and REDDIT-B

| Dataset | Class | Generated Examples | | |
|---|---|---|---|---|
| IMDB-B | Class0 (Action) | | | |
| | Class1 (Romance) | | | |
| Reddit-B | Class0 (Question/ Answer) | | | |
| | Class1 (Discussion) | | | |

Figure 9: Simpler Explanations of IMDB-Binary and REDDIT-Binary

Figure 9 shows some smaller sized explanations generated by Graphon-Explainer on the IMDB-Binary and Reddit-Binary datasets.

## J    Comparative Results with other Metrics and Methods

In this section, we compare the performance of Graphon-Explainer with GDM(Nian et al., 2024) on three datasets. We report class score averages for 10 explanations generated per class on the dataset as a whole as reported by the authors of GDM in Table 6. Further, we also adopt a metric proposed by the authors called model fidelity. Model fidelity demonstrates the usefulness of the generated explanations by training a surrogate model on the explanations having a similar structure as the classifier and measuring the ratio of cases on the test set where the surrogate model gives the same decision as the original classifier. As shown in Table 6 and Table 7 Graphon-Explainer performs better or equally on all except on one metric on one dataset. Figure 10 also shows the explanations and graphons generated by Graphon-Explainer on the SHAPE dataset.

Table 6: Class Scores

| Dataset | GDM | Graphon-Explainer |
|---|---|---|
| **MUTAG** | $0.8267 \pm 0.0470$ | $1.0 \pm 0.00$ |
| **BA2Motif** | $1.0 \pm 0.0000$ | $1.0 \pm 0.00$ |
| **SHAPE** | $1.0 \pm 0.0000$ | $1.0 \pm 0.00$ |

Table 7: Model Fidelity

| Dataset | GDM | Graphon-Explainer |
|---|---|---|
| **MUTAG** | $94.73 \pm 0.00$ | $96.81 \pm 0.005$ |
| **BA2Motif** | $98.00 \pm 0.00$ | $96.19 \pm 0.034$ |
| **SHAPE** | $84.00 \pm 8.00$ | $92.41 \pm 0.143$ |

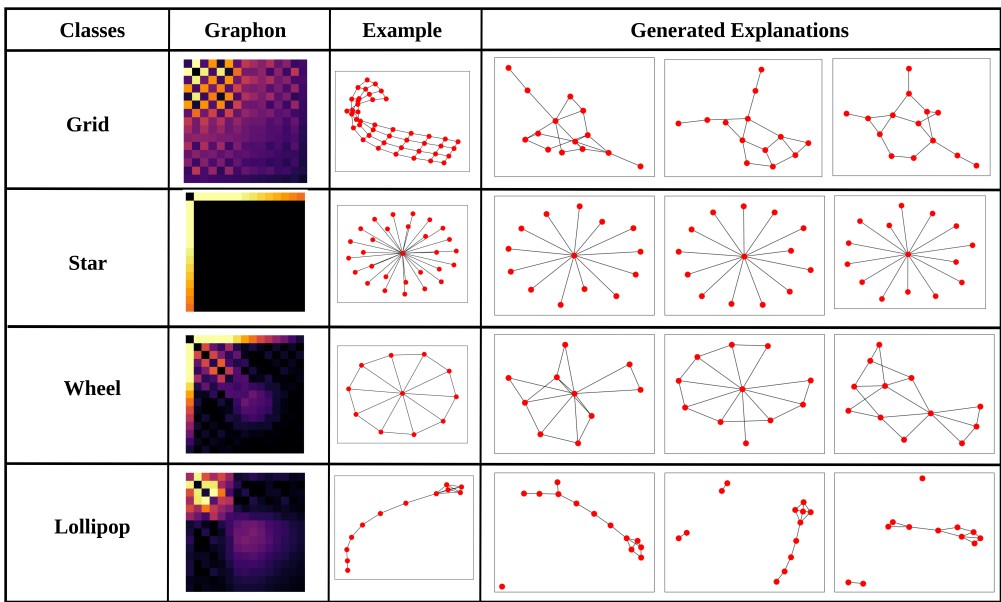

Figure 10: Explanations on SHAPE Dataset generated by Graphon-Explainer

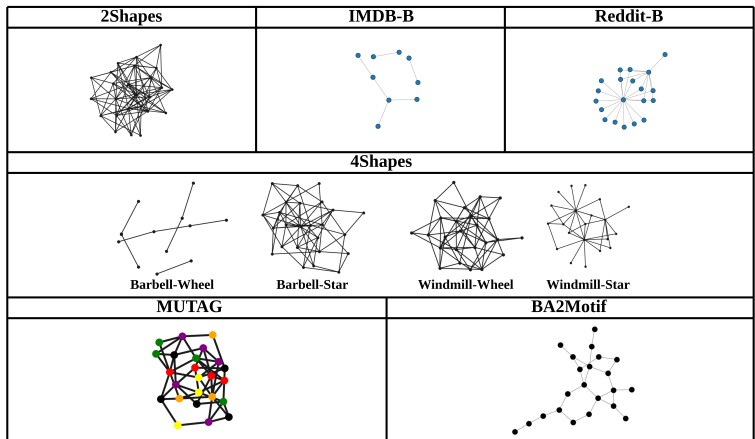

Figure 11: Visual Results of GNNBoundary

## K   Visual Results of GNNBoundary

Figure 11 demonstrates the motifs generated by GNNBoundary on all datasets for decision boundaries between different target classes.

## L   Sensitivity Analysis

### L.1   Setting number of samples on the dataset

The following is the hyperparameter setting for Graphon-Explainer used on the corresponding datasets.

Table 8: Hyperparameter values of $n_{samples}$ on all datasets

| Dataset | #samples for class explanation | #samples for boundary explanation |
|---|---|---|
| 2Shapes | 3 | 20 |
| BA2Motif | 10 | 50 |
| 4Shapes | 4 | 800 |
| IMDB-B | 50 | 200 |
| Reddit-B | 10 | 150 |
| MUTAG | 20 | 100 |

### L.2   Sensitivity Analysis on $\lambda$ and Number of Nodes

Figure 12 outlines the variation in class scores for the explanation graphs with the number of nodes for all the target classes and with varying mixup parameter $\lambda$ on the boundary classes. On all the datasets for scores reported in Table 2, we choose $K$ to be the median number of nodes in the dataset and $\lambda$ is varied from 0.4 to 0.6 for generation of the boundary motifs, and the $\lambda$ yielding the best score is chosen.

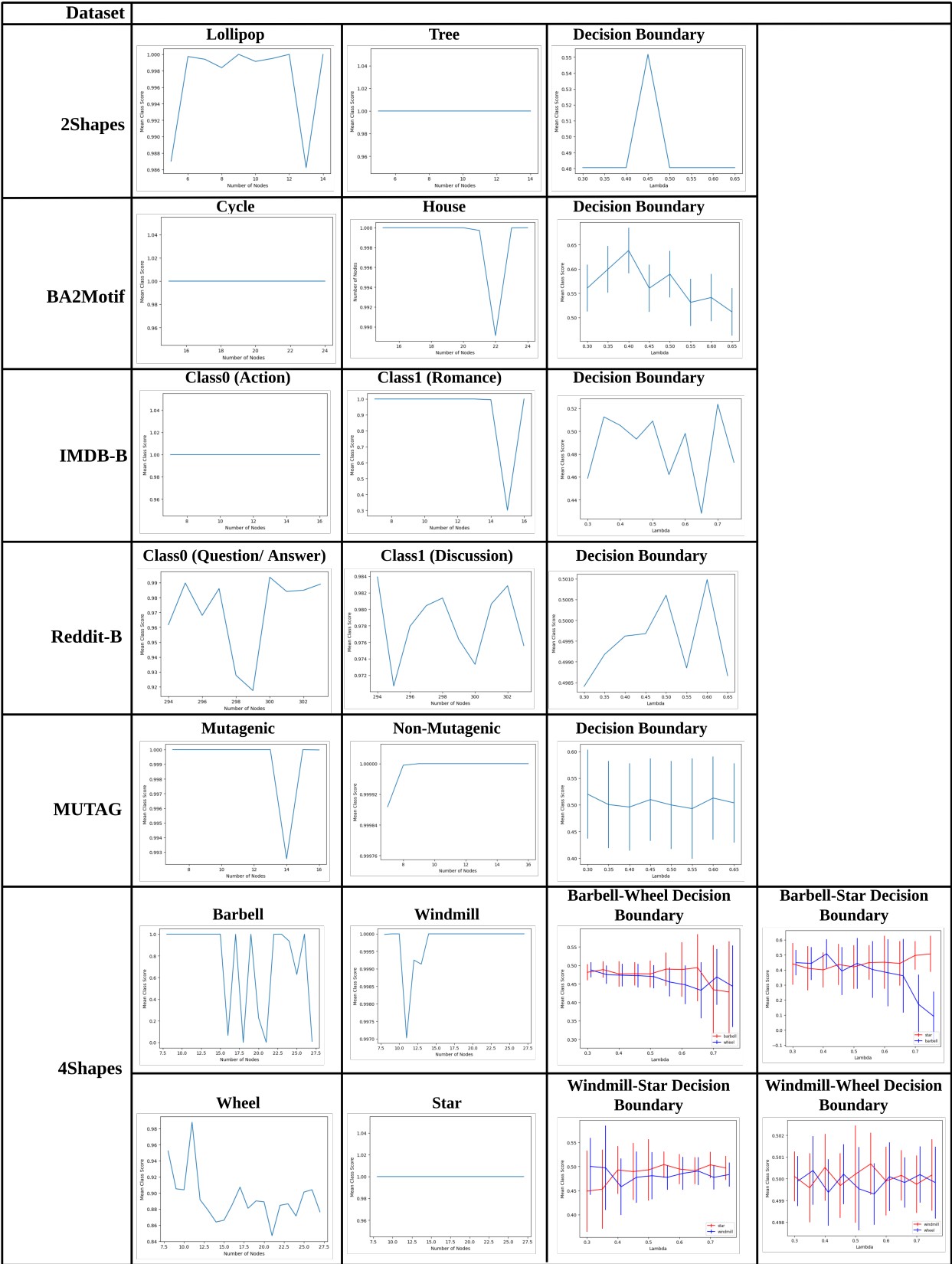

Figure 12: Sensitivity Analysis on all Datasets

