# OpenReview forum: "Graphon-Explainer: Generating Model-Level Explanations for Graph Neural Networks using Graphons"
_TMLR — Accepted by TMLR_

### Review · Reviewer_wLSB · 2024-09-06

**Summary Of Contributions:**

Authors propose a method to study decision boundary and flat region of loss function minimum for a GNN-based classifier. Graph data is heterogeneous and inherently discrete, so classical gradient-based methods used in other domains (e.g., in CV see https://arxiv.org/abs/1412.1897 for discussion of flat region with small loss, see https://arxiv.org/abs/1312.6199 for the study of decision boundary) are not directly applicable. Authors overcome this issue by using techniques proposed in https://arxiv.org/abs/2202.07179. More specifically, they summarize statistics about graphs in each class with graphons and use them to study both regions with small loss and decision boundary (interpolation of learned features is also used in the later case). Authors perform a series of experiments with synthetic and real-world data. The results demonstrate that the developed method is superior to other recent interpretability techniques.

**Audience:**

Yes

**Claims And Evidence:**

Yes

**Requested Changes:**

1. What if the reader does not know what graphon is? Consider explaining this briefly in the abstract.
2. GNNs can be used in many problems including, for example, the solution of PDEs https://arxiv.org/abs/2202.03376. The constructions discussed in the article are only applicable to the classification. On the other hand the title implies that the method can be used for general GNN. I kindly ask authors to resolve this issue by providing additional discussion about that in the abstrac or the main body of the paper.
3. Graphons are further discussed in section 2 without a definition which appears in the next section. This makes it harder for the reader who did not know what graphon is to digest the content of the paper. I suggest authors explain graphons earlier.
4. In section 4.1. the problem setup involves maximization over graph $G$. This is not a well-poised problem since authors do not provide the space of graphs $G$ belongs to. From later text, I can guess authors assume there is some unknown underlying distribution $p(G, C)$. A related conditional distribution $p(G|C)$ is approximated from data using graphon estimation (+ extension for feature estimation). Since authors optimize over samples from this estimated distribution, I assume in section 4.1 one should also specify that $G$ are drawn from $p(G|C)$.
5. It is not clear how features are mixed and how graphons are estimated. I also consulted the G-Mixup paper https://arxiv.org/abs/2202.07179 which contains a similar vague description. Can the authors please provide a more detailed explanation on graphons and feature mixing? In particular, I have the following question:
   1. Authors write “Algorithm 4 describes the method we use to approximate graphons.” Unfortunately, Algorithm 4 does not help since it contains “graphon estimator g” as an input. In essence Algorithm 4 is a suggestion to use “graphon estimator” to estimate graphon. Can the authors please explain clearly, preferably in the form of pseudocode or concrete mathematical expression, what one should do with sorted adjacency matrices?
   2. Related to the previous question, the description in Section 4.3. is not helpful. Authors write “We use the USVT method (Chatterjee, 2015) to estimate the step function. This method aligns the nodes of the graph in each dataset based on certain node properties and then proceeds to estimate the step function from the aligned adjacency matrices.” From this sentence I can infer that to estimate step function I should estimate it from aligned adjacency matrices. How to do this is not detailed. Since graphons are central to the method advocated in the paper I think it is instructive to include this algorithm in the main text or appendix.
   3. In a similar way it is not clear how features are generated. Authors provide a reference to https://arxiv.org/abs/2202.07179 where this part is explained in precisely the same language without clear mathematical or algorithmic description. Suppose I have a set of graphs $G_1, \dots, G_N$ with node features $f_1, \dots, f_N$. What should I do with node features to obtain features of graphs, generated from estimated graphon?
   4. Graphon is a function [0, 1]^2 -> [0, 1] with certain properties. What do authors mean by “graphon nodes”? Functions do not have nodes. Perhaps, it is related to interpretation of graphon as a “graph with infinite number of nodes”. Can the authors please clarify this part?
6. Section 4.5. that describes how to generate graphs near the decision boundary. For that authors first propose (for two selected classes) to extract learned features from the layer preceding the softmax function, mix these features and track predicted classes based on the mixing parameter. Based on that one can conclude whether two classes have a shared decision boundary. I have several question:
   1. In this procedure one uses synthetic data obtained from graphons or the original data?
   2. Authors write in the introduction “... it [Graphon-Explainer] does not need access to training data embeddings to generate explanations.”  Regardless of the answer to the previous question, Graphon-Explainer directly uses embeddings, so I suggest rephrasing or clarifying the sentence from the introduction.
7. How the number of nodes is selected when synthetic graphs are generated with estimated graphon ($N$ in equation (4))?
8. In the theoretical analysis authors focus on graph motifs. As I understand this section ignores the part that nodes of the graph have features. Can the features be included in the theoretical analysis?
9. Authors claim generated graphs include discriminative motifs. This motif is available only indirectly since we know it is included in generated graphs with high classification score. Is there a way to directly find this motif?
10. Is the generation of new samples necessary for finding a model-level explanation? If one selects samples from the dataset with a largest classification score they also should contain discriminative motifs. Can the authors report the difference between classification scores and densities for best samples from train/test sets and samples generated by their method?
11. In the definition of boundary margins authors need to clarify what “graph belongs to class $c_1$” means. Since synthetic data is used is it not clear how this class is defined. If it is defined with the classifier itself, authors need to specify the decision rule used, e.g., the usual one is the class with largest predicted probability.
12. In boundary thickness $G_b$ should probably be replaced with $G_{c_2}$.

In addition to all technical problems highlighted above, I would like to ask authors to provide general comments on the goals of the method. More specifically, it is not clear to me whether the search for “discriminative motif” helps in interpretability. One may argue that in all cases authors studied (table 3) the graph found for each class is not easier to analyze than any graph from the same class from the training set. How precisely generated graphs can help the user in the analysis of the classifier? This question is more or less clear for the decision boundary since there are quantitative measures of how complex the boundary is and how well classes are separated. For the case of individual classes. class score does not seem to be very informative and the density only refers to the complexity of the generated graph, so the quality of the method, not the classifier or the problem. Can the authors expand discussion on why the generated “discriminative motif” is useful?

**Strengths And Weaknesses:**

I find overall that the paper is well-written and most concepts are reasonably explained (see below for exceptions). The discussion of performed experiments is extensive, and the results are presented with many helpful illustrations.

I also find that several important explanations regarding the graphon estimation are lacking. Besides that, it is not clear to me whether the obtained explanation actually entails interpretability of GNN. I expand on these topics in the next section of review.

---

> ### Author Response · Authors · 2024-09-11
> **Reply to Reviewer wLSB**
>
> We thank the reviewer for a very thorough reading of our paper and raising many interesting points and questions that has resulted in a much better revised version of our paper. Next, we try to answer all the queries and address the concerns that the reviewer raised.
>
> 1) What if the reader does not know what graphon is? Consider explaining this briefly in the abstract.
>
> **Response**
>
> We have added a brief explanation of graphon in the abstract.
>
> 2) GNNs can be used in many problems including, for example, the solution of PDEs https://arxiv.org/abs/2202.03376 (https://arxiv.org/abs/2202.03376). The constructions discussed in the article are only applicable to the classification. On the other hand the title implies that the method can be used for general GNN. I kindly ask authors to resolve this issue by providing additional discussion about that in the abstrac or the main body of the paper.
>
> **Response**
>
> We have revised our abstract to include that our method aims to provide explanations on the task of graph classification.
>
> 3)  Graphons are further discussed in section 2 without a definition which appears in the next section. This makes it harder for the reader who did not know what graphon is to digest the content of the paper. I suggest authors explain graphons earlier.
>
> **Response**
>
> We have moved the Preliminaries section before the Related Work section so that the definition of graphons can be found earlier.
>
> 4) In section 4.1. the problem setup involves maximization over graph $G$. This is not a well-poised problem since authors do not provide the space of graphs $G$ belongs to. From later text, I can guess authors assume there is some unknown underlying distribution $p(G,C)$. 5. A related conditional distribution $p(G|C)$ is approximated from data using graphon estimation (+ extension for feature estimation). 6. Since authors optimize over samples from this estimated distribution, I assume in section 4.1 one should also specify that $G$ are drawn from $p(G|C)$.
>
> **Response**
>
> Thank you for pointing this out. We have revised Section 4.1 to specify the distributional assumptions of our method.
>
> 5) It is not clear how features are mixed and how graphons are estimated. I also consulted the G-Mixup paper https://arxiv.org/abs/2202.07179 (https://arxiv.org/abs/2202.07179) which contains a similar vague description.
> Can the authors please provide a more detailed explanation on graphons and feature mixing? In particular, I have
> the following question:
> 1. Authors write “Algorithm 4 describes the method we use to approximate graphons.” Unfortunately, Algorithm 4 does not help since it contains “graphon estimator g” as an input. In essence Algorithm 4 is a suggestion to use “graphon estimator” to estimate graphon. Can the authors please explain clearly, preferably in the form of pseudocode or concrete mathematical expression, what one should do with sorted adjacency matrices?
>
> 2. Related to the previous question, the description in Section 4.3. is not helpful. Authors write “We use the USVT
> method (Chatterjee, 2015) to estimate the step function. This method aligns the nodes of the graph in each dataset based on certain node properties and then proceeds to estimate the step function from the aligned
> adjacency matrices.” From this sentence I can infer that to estimate step function I should estimate it from aligned adjacency matrices. How to do this is not detailed. Since graphons are central to the method advocated in the paper I think it is instructive to include this algorithm in the main text or appendix.
> 3. In a similar way it is not clear how features are generated. Authors provide a reference to
> https://arxiv.org/abs/2202.07179 (https://arxiv.org/abs/2202.07179) where this part is explained in precisely the same language without clear mathematical or algorithmic description. Suppose I have a set of graphs $G_1. \cdots, G_N$
> with node features $f_1, \cdots, f_N$. What should I do with node features to obtain features of graphs, generated from estimated graphon?
> 4. Graphon is a function $[0, 1]^2 -> [0, 1]$ with certain properties. What do authors mean by “graphon nodes”? Functions do not have nodes. Perhaps, it is related to interpretation of graphon as a “graph with infinite number of nodes”. Can the authors please clarify this part?
>
> **Response**
>
> We have revised Algorithm 4 to detail the method of estimating the graphon and sampling a graph from the graphon. We have also revised the description of generating the node feature vectors within the text in Section 4.4 and 4.5 to incorporate as faithfully as possible in text the way to generate node feature vectors as described in Algorithm 4.
> Actually by graphon nodes we meant the nodes of the graph sampled from the graphon, however we appreciate the difficulty in understanding this for any reader and hence, we have revised the corresponding description in Section 4.4.

---

> > ### Author Response · Authors · 2024-09-11
> > **Reply to Reviewer wLSB (Continued)**
> >
> > 6) Section 4.5. that describes how to generate graphs near the decision boundary. For that authors first propose (for two selected classes) to extract learned features from the layer preceding the softmax function, mix these features
> > and track predicted classes based on the mixing parameter. Based on that one can conclude whether two classes have a shared decision boundary. I have several question:
> > 1. In this procedure one uses synthetic data obtained from graphons or the original data?
> > 2. Authors write in the introduction “... it [Graphon-Explainer] does not need access to training data embeddings to generate explanations.” Regardless of the answer to the previous question, Graphon-Explainer directly uses embeddings, so I suggest rephrasing or clarifying the sentence from the introduction.
> >
> > **Response**
> >
> > Yes, essentially we sample two graphs that were classified to two different target classes and interpolate between their embeddings (note that these embeddings are synthetically generated since they are linear combinations of embeddings of these two graphs) which can be thought of as walking in a straight line in the embedding space, all the while checking if we need to pass through the decision region of another class except these two target classes.
> > You are indeed correct, that Graphon-Explainer directly uses embeddings to compute adjacency score of two classes. However, the distinction with GNNInterpreter is that Graphon-Explainer only accesses the embeddings of the query dataset which the user seeking explanation posseses whereas GNNInterpreter needs access to training data embeddings as a part of the objective it optimizes to generate explanations. In lieu of this, we have rephrased the corresponding sentence in the introduction as " Similarly, unlike GNNInterpreter , which requires access to training data embeddings to generate explanations, Graphon-Explainer only requires access to the embeddings of the query dataset that the user possesses, while seeking to explain the model's behavior."
> >
> > 7)  How the number of nodes is selected when synthetic graphs are generated with estimated graphon ($N$ in equation (4))?
> >
> > **Response**
> >
> > Note that, equation 4 describes how to sample a graph from a true graphon function. However, since it is impossible to know the true graphon function(as it occurs in the limit when the number of nodes go to infinity), we approximate the true graphon using  step function methods as described in previous literature. As discussed in the Algorithm 4 of the revised version, $N$ is fixed in the graphon estimation phase itself while deciding the number of partitions on which to approximate the step function, hence the obtained approximated graphon $S^c$ is a $N\times N$ matrix, with entries between 0 and 1 from where we sample the adjacency matrix using each entry as a Bernoulli weight as described in Equation 4. Note that  the obtained $S^c$ can be viewed as a graphon function when the unit interval is partitioned into $N$ partitions $\{P_1,..,P_N\}$ where $S^c(p_i,p_j)= S^c_{ij}$ where $p_i \in P_i, p_j \in P_j$.  Hence, graphs of smaller nodes are generated using step function approximation on lesser partitions, which are a coarser approximation of the true graphon function and they might not be able to generate graphs that faithfully reflect the the topological patterns that the classifier learns when the sample graphs have large number of nodes like in the case of the REDDIT dataset. Keeping this information bottleneck in mind, we choose $N$ to be the median number of nodes in the dataset for results reported in Table 2 and Table 3. It should also be noted here that prior works such as GNNInterpreter fix the maximum number of nodes that their generative model can generate and employ a budget term in their loss function term  to control for the number of nodes that the explanations contain. We could have adopted a similar approach here by estimating the step function using the maximum number of nodes and sampling $N$ nodes from the step function using equation 4 to sample a N node graph, however the theory on estimation of graphons guarantees a better approximation of the graph topology when the estimation process itself is tailored to size of the graph one wants to generate. Infact, the sensitivity analysis in the appendix substantiates this claim by showing that class scores do not vary much with change in the number of nodes.

---

> ### Author Response · Authors · 2024-09-11
> **Reply to Reviewer wLSB (Continued)**
>
> 8) In the theoretical analysis authors focus on graph motifs. As I understand this section ignores the part that nodes of the graph have features. Can the features be included in the theoretical analysis?
>
> **Response**
>
> The  theoretical analysis  indeed focuses on the graph structure rather than on the node features primarily due to two reasons. Firstly, the task at hand is graph classification for which the goal of a GNN as mentioned in Fox \& Ramanchandran (https://arxiv.org/pdf/1912.10206) is to minimize structural noise, hence we investigate in greater detail how GNNs understand structure and provide theoretical guarantees of our method based on it. Secondly, many basic GNN variants which are commonly used such as GIN, GCN, GraphSAGE were developed with the motivation of encoding the graph topology in the best way possible to perform underlying tasks. A notable phenomenon was observed when generating explanations for the Mutagenic class in the MUTAG dataset. A key characteristic of this class is the presence of fused carbon rings. Interestingly, even when these fused rings contained atoms other than carbon, they still received a class score of 1, indicating that the classifier prioritizes the fused ring structure over the specific constituent atoms
> Further, we would like to emphasise that to the best of our knowledge this is the only work in model-level explainability of GNN that gives theoretical guarantees on the generated explanations. Having said that, it is indeed a limitation of the theoretical analysis of our method that in its current form it cannot incorporate the node feature matrix.
>
> 9) Authors claim generated graphs include discriminative motifs. This motif is available only indirectly since we know it is included in generated graphs with high classification score. Is there a way to directly find this motif?
>
> **Response**
>
> No, as noted by previous works such as GNNInterpreter, D4Explainer there is no way to directly find the exact graph patterns that a classifier has learnt to identify a target class, hence we need to rely on a generative mechanism to generate such graph patterns.
>
> 10) Is the generation of new samples necessary for finding a model-level explanation? If one selects samples from the dataset with a largest classification score they also should contain discriminative motifs. Can the authors report the difference between classification scores and densities for best samples from train/test sets and samples generated by their method?
>
> **Response**
>
> The reviewer has raised a very interesting point. To answer this question, we would like the reviewer to refer to Figure 3a in the manuscript and see the explanations generated for the Windmill and Wheel classes on the 4Shapes dataset. We can see a motif containing central hub node with spokes with minor variations for both the explanations. Note that, this motif is common to several instances of the class in question. Thus, this explanation has a greater generalization capacity and explains the basis on which the classifier identifies the Windmill/Wheel class as a whole which is a central goal of model-level explanation.  Secondly, the similarity of explanations of both the classes shows that the classifier bases its decision of identifying Windmills/Wheels on  similar motifs, hence the classifier can potentially misclassify these classes. This finding is actually validated by the confusion matrix of the classifier. Even if samples from the dataset belonging to these target classes have a class score of 1, it would be impossible to gain insight into why the classifier misclassifies these two classes without the generated explanations. Also as noted in the answer to a previous question, on the MUTAG dataset we see that the classifier identifies mutagenic class based on the fused ring structure irrespective of the constituent atoms of the fused rings. This insight that the structure matters more than the constituent atoms would be impossible to understand using samples from the dataset as they would all contain only fused carbon rings.
> Further, after this answer if the reviewer still deems it necessary to report the difference in class scores and density of the best sample and the generated explanation, we would be more than happy to oblige.
>
> 11) In the definition of boundary margins authors need to clarify what “graph belongs to class $c_1$” means. Since synthetic data is used is it not clear how this class is defined. If it is defined with the classifier itself, authors need to specify the decision rule used, e.g., the usual one is the class with largest predicted probability.
>
> **Response**
>
>  Yes, the decision rule is the usual exact rule you mentioned. We have clarified it in the definition like this "$G_{c_1}$ belongs to class $c_1$ by having the highest class score for class $c_1$ according to $f$".

---

> ### Author Response · Authors · 2024-09-11
> **Reply to Reviewer wLSB (Continued)**
>
> 12)  In boundary thickness $G_b$ should probably be replaced with $G_{c_2}$.
>
> **Response**
>
> We would like to respectfully disagree with the reviewer on this. Boundary thickness is an asymmetric measure which is different for classes $c_1$ and $c_2$ when they share a decision boundary.
>
> **General Comments**
>
> In addition to all technical problems highlighted above, I would like to ask authors to provide general comments on the goals of the method. More specifically, it is not clear to me whether the search for “discriminative motif” helps in interpretability. One may argue that in all cases authors studied (table 3) the graph found for each class is not easier to analyze than any graph from the same class from the training set. How precisely generated graphs can help the user in the analysis of the classifier? This question is more or less clear for the decision boundary since there are quantitative measures of how complex the boundary is and how well classes are separated. For the case of individual classes. class score does not seem to be very informative and the density only refers to the complexity of the generated graph, so the quality of the method, not the classifier or the problem. Can the authors expand discussion on why the generated “discriminative motif” is useful?
>
> **Response**
>
> Note that, the primary goal of proposing this method was to shed light on the working of a GNN classifier by providing a unified understanding of the motifs it uses to distinguish a target class and also shed light on its decision boundary properties. To the best of our knowledge, this is the first method to provide such an unified understanding and we proposed this method because we believe each component in isolation provides an incomplete characterization of the classifier, hence we request the reviewers to keep this vital point in mind while analysing our results.\\
> We also agree with the reviewer that class score as a metric of understanding the classifier is non-informative to the user. Indeed, if maximizing class score was the only goal of explanations on individual target classes, we could use samples from the dataset itself which have high class scores as explanations. But as I have hopefully pointed out in the answer to your previous questions, like in the case of Windmill-Wheel identification where the common structure of the explanations explains why the classifier misclassifies these two classes and the basis of mutagenic class identification  is the fused ring structure irrespective of the constituent atoms,  that there is a larger goal of identifying  discriminative motifs that the classifier has learnt which illustrates the general basis of how the classifier identifies several instances of a class and since there is no way to directly find this motif, the only way is to maximize the class score. We further add experiments (in Appendix H) showing how differently two classifiers understand the same class depending on their inductive biases even when they have a perfect accuracy on a class, informing an user about which classifier to select even when they have a perfect accuracy. Lastly, we also supplement experiments (in Appendix J) to show that an user can train another surrogate model using the obtained motif explanations which performs with almost the same accuracy as the original classifier. Since the surrogate model is trained on the explanations, it is guaranteed to ground its decisions of identifying a class on the patterns illustrated by the explanations.
>
> We would be happy to further address any remaining concerns you have.

---

### Review · Reviewer_1V5r · 2024-09-06

**Summary Of Contributions:**

This paper proposes Graphon-Explainer, which explains discriminative features of graph neural networks (GNNs) by learning a class-conditional generative model parametrized by graphons. Specifically, given a dataset of graphs and corresponding GNN classification labels, a graphon is learned to approximate the collection of graphs belonging to each class. Graphs near the GNN decision boundary can be generated by sampling from a linear interpolation of graphons of distinct classes. The authors verify that Graphon-Explainer outperform baselines GNNInterpreter and XGNN on a variety of datasets.

**Audience:**

Yes

**Broader Impact Concerns:**

None.

**Claims And Evidence:**

No

**Requested Changes:**

**Critical**

- [W1] : I am not sure what change would address [W1]. Unless my understanding of the method is wrong, the authors would have to modify the method such that it is able to explain discriminative features specific to individual GNNs. After making such a change to the method, the authors could demonstrate how the method is able to generate different explanations for different GNNs.

**Not Critical**

- [W2] : The authors could add a table, as mentioned in Strengths and Weaknesses.

**Strengths And Weaknesses:**

**Strengths**

- [S1] The writing is very clearly presented, and I had no problem following the exposition.
- [S2] Due to conceptual simplicity of the method, the authors are able to provide theoretical guarantees using graph-theoretical results.
- [S3] The authors provide a thorough experimental comparison of the proposed and baseline methods.

**Weaknesses**

- [W1] There is a critical flaw in the proposed method, that in some scenarios, the method explains the data, not the model. For instance, suppose that there are two GNNs $f_1$ and $f_2$ which have perfect accuracy, but use different discriminative features to classify inputs.
 Graphon-Explainer, would yield identical class-conditional graphons for both $f_1$ and $f_2$ as they are learned using identical datasets. Thus, in scenarios where we would like to compare two or more highly accurate GNNs, Graphon-Explainer would not be really useful.
- [W2] I found the description of the baselines to be slightly unclear. It would be beneficial to have a table comparing which methods need access to training data, GNN internal components, etc.

---

> ### Comment · Reviewer_81Da · 2024-09-08
>
> Congratulations on identifying W1!
>
> If I understood the method correctly, I agree with you. I believe W1 is an important issue that needs to be clarified.
>
> Best,
> Reviewer 81Da

---

> > ### Comment · Reviewer_wLSB · 2024-09-08
> >
> > W1 is not completely fair, because authors did not use graphons per se to explain the model.
> >
> > Objects that authors use for interpretability are directly related to models under scrutiny since they perform optimisations described by equations (2) and (3) that are explicitly model-dependent. So, even when the models produce identical predictions for each class, samples generated for small flat loss regions and on the decision boundary can potentially be different for each model.
> >
> > I suspect that not much will change in practice if authors use ground truth classes in place of model-generated. It seems to me that the main end of graphons is, in some sense, to provide smooth structure to the discrete problem.
> >
> > That said, it would be interesting to see an explicit example that demonstrates differences induced by inductive biases of the models, not by the graphons used to model the data.

---

> ### Author Response · Authors · 2024-09-11
> **Reply to Reviewer 1V5r**
>
> We thank the reviewer for the feedback and raising a very interesting point that we elaborate on below. We are happy that the reviewer has found our writing clear and our experiment section extensive.
>
> [W1]  There is a critical flaw in the proposed method, that in some scenarios, the method explains the data, not the model. For instance, suppose that there are two GNNs $f_1$ and $f_2$ which have perfect accuracy, but use different discriminative features to classify inputs. Graphon-Explainer, would yield identical class-conditional graphons for both and as they are learned using identical datasets. Thus, in scenarios where we would like to compare two or more highly accurate GNNs, Graphon-Explainer would not be really useful.
>
> **Response**
>
> First of all, we thank the reviewer of raising this point which has resulted in a great discussion. We answer this question part by part:
> Firstly, as reviewer wLSB has rightly pointed out, we do not sample from the learned graphons randomly to generate explanations on individual target classes or motifs near the decision boundary. Instead, as mentioned in Algorithm 1, the generation of the explanations are wholly guided by the classifier as we optimize scores in Equation 2 and Equation 3  to generate explanations. So, in the scenario as reviewer 1V5r describes, where two perfect classifiers are employed to classify the dataset, the learned graphons would be the same(as correctly pointed out by the reviewer), however the samples which would be selected as explanations by optimizing equation 2 for a target class would only be the same if both the classifiers have learnt the same discriminative motifs for a particular target class. Particularly, three possible scenarios can arise when a two classifiers are employed on the same dataset:
>
> a) both classifiers classify a target class based on exactly the same set of discriminative motifs in which case Graphon-Explainer would provide the same explanations for both as optimizing equation 2 would yield the same set of $G^*$ graphs
>
> b) the set of discriminative motifs that the classifiers learn for a target class have no overlap in which case, Graphon-Explainer would provide different discriminative motifs as we would reach different $G^*$ by optimizing equation 2.
>
> c) the set of discriminative motifs that the classifiers learn  for a target class is overlapping to a certain degree in which case optimizing equation 2 may or may not lead to the same $G^*$, hence in this case Graphon-Explainer may or may not produce same explanations.
>
> Here, we would also like to comment on our usage of graphons as generative models. As pointed out rightly by reviewer wLSB, one reasoning behind using graphons is to come up with a way of smoothly approximating the topology of a set of graphs. Further, the learnt graphon for each target class when used as a generative model serves as an anchor which does not let the generated explanations stray far away from the data distribution. Since, any neural network in general and a GNN in particular is a non-invertible many to one function, optimizing equation 2 or 3 without any distributional assumptions would lead to strange solutions( like a single node in the case of many datasets for XGNN as shown in Figure 6) with high class scores, which cannot serve as explanations. Hence, using graphons as the generative model restricts the space within which the objective is maximized. Generating in-distribution explanations as a goal is important and many methods have been proposed even in this year to tackle it.(see [1] and [2])
>
> Secondly, we would also like to mention here that our method is well-equipped to find out, using graph embeddings of the GNN, which classes share a decision boundary and examine different properties of such decision boundaries. It should be noted here that Algorithm 5 which describes how to compute the adjacency score between two classes is exclusively dependent on how the classifier embeds the graphs in the dataset. Note that, it cannot be guaranteed that two perfect classifiers would embed the graphs in exactly the same way, hence, the two classes that share a boundary for one classifier might not share a decision boundary when the other classifier is used. Note that, in this case the boundary graphons even would not be the same. Even in the case that the same classes share a decision boundary the boundary metric values (which are again exclusively dependent on the classifier embeddings) may be different for both the classifiers. Lastly, the generated motifs on the decision boundary is selected by optimizing equation 3 , hence the motif that is selected is again dependent only on the classifier.

---

> ### Author Response · Authors · 2024-09-11
> **Reply to Reviewer 1V5r (Continued)**
>
> [W1]  **Response Continued**
>
> Finally, we conduct an experiment( added in Appendix H) on the 4Shapes dataset using a GCN and a GIN architecture, both of which perform perfectly on the same target classes. Even then, we elucidate how different the explanations on both architectures are on the Barbell class, even though the graphons are the same since both the classifiers have a 100\% accuracy on this class.  Further, we also elucidate that for the GIN architecture different classes share decision boundaries than for the GCN architecture and even when same classes share decision boundaries they have different properties.
>
> We hope this answer clarifies the main concern that the reviewer has about our method.
>
> [W2] I found the description of the baselines to be slightly unclear. It would be beneficial to have a table comparing which methods need access to training data, GNN internal components, etc.
>
> **Response**
>
> Yes, we promise to add this table in the next revised version of our paper.
>
> **References**
>
> 1) Generating In-Distribution Proxy Graphs for Explaining Graph Neural Networks Zhuomin Chen, Jiaxing Zhang, Jingchao Ni, Xiaoting Li, Yuchen Bian, Md Mezbahul Islam, Ananda Mondal, Hua Wei, Dongsheng Luo Proceedings of the 41st International Conference on Machine Learning, PMLR 235:7712-7730, 2024
>
> 2) D4explainer: In-distribution explanations of graph neural network via discrete denoising diffusion  Chen, Jialin and Wu, Shirley and Gupta, Abhijit and Ying, Rex Advances in Neural Information Processing Systems, 2024
>
> We would be happy to address any further concerns that you may have about our work.

---

> ### Author Response · Authors · 2024-09-12
> **Reply to Comment by Reviewer 81Da**
>
> We request you to refer to our answer on W1 to reviewer 1V5r  for a detailed clarification of this issue. Further, please see Appendix H which demonstrates how two perfectly accurate models have very different explanations on the same target class which can be generated using our method. This experiment also demonstrates that our method is also able to diagnose the different decision boundary properties of the two models.

---

> ### Author Response · Authors · 2024-09-12
> **Reply to Comment by Reviewer wLSB**
>
> It seems to me that the main end of graphons is, in some sense, to provide smooth structure to the discrete problem.
>
>
> **Response**
>
>
> Kindly refer to our answer(specifically para 2) to W1 of reviewer 1V5r for our clarification on the use of graphons. In addition to the reasons we give in that answer on the usage of graphons,  we would also like to mention that graphons have a regular structure unlike  graphs due to which it is possible to mix them to generate boundary motifs naturally.
>
> It would be interesting to see an explicit example that demonstrates differences induced by inductive biases of the models.
>
> **Response**
>
> Kindly see Appendix H of our revised paper where we conduct an experiment to demonstrate how different inductive biases of two different models lead to different explanations of the same target class.

---

### Review · Reviewer_81Da · 2024-09-08

**Summary Of Contributions:**

The authors propose a novel method-level post-hoc explainer with a straightforward approach: learning a graphon for each target class. They conclude by evaluating their method on three real-world datasets and three synthetic ones.

**Audience:**

Yes

**Broader Impact Concerns:**

Given the complexity of the extracted real-world explanations, I find it difficult to see the potential impact of the proposed method.

**Claims And Evidence:**

No

**Requested Changes:**

The authors should compare their method's performance against other relevant works. Additionally, they should provide a more detailed discussion on the expected explanations in potential real-world applications. From an interpretability standpoint, it is difficult to discern meaningful insights from large graphs, such as those in Figure 3b (IMDB and Reddit).

**Strengths And Weaknesses:**

Strengths
S1: The method is intuitive and easy to follow.
S2: The results are promising.

Weaknesses

W1: The author suggests that "multiple distinct $G^*$ may exist that maximize equation (2)." However, they do not test this claim on publicly available multi-substructure datasets. (i.e. https://pytorch-geometric.readthedocs.io/en/latest/generated/torch_geometric.datasets.BAMultiShapesDataset.html#torch_geometric.datasets.BAMultiShapesDataset)

W2: Other works in GNN global-explanability are not cited [1][2][3][4].

W3: Upon reviewing the experimental results (Figure 3b IMDB and reddit), it remains unclear what specific insights or explanations can be derived from the generated subgraph.

W4:It is unclear how node and edge features are handled by the graphon.



[1]Nian, Yi, et al. "Globally Interpretable Graph Learning via Distribution Matching." Proceedings of the ACM on Web Conference 2024. 2024. [2] Müller, Peter, et al. "GraphChef: Decision-Tree Recipes to Explain Graph Neural Networks." The Twelfth International Conference on Learning Representations. 2024. [3] Azzolin, Steve, et al. "Global Explainability of GNNs via Logic Combination of Learned Concepts." The Eleventh International Conference on Learning Representations. [4] Yu, Zhaoning, and Hongyang Gao. "MAGE: Model-Level Graph Neural Networks Explanations via Motif-based Graph Generation." arXiv preprint arXiv:2405.12519 (2024).

---

> ### Author Response · Authors · 2024-09-11
> **Reply to Reviewer 81Da**
>
> We thank the reviewer for the valuable feedback on our paper and pointing us to more literature on GNN explanations. We are happy that the reviewer has found our method intuitive and our results promising.  Next, we try to address all the concerns that the reviewer has raised.
>
> W1: The author suggests that "multiple distinct may exist that maximize equation (2)." However, they do not test this claim on publicly available multi-substructure datasets. (i.e. https://pytorch-geometric.readthedocs.io/en/latest/generated/torch_geometric.datasets.BAMultiShapesDataset.html#torch_geometric.datasets.BAMultiShapesDataset)
>
> **Response**
>
> The authors would like to kindly clarify that this statement which was made with reference to equation 2 only meant that depending on the functional form of $f(.)$ which is the GNN classifier, there might exist multiple discriminative motifs that attain the maximum class score for a particular target class $c$. We made this statement explicitly, lest a reader should mistakenly believe that $G^*$ is an unique motif, which  Graphon-Explainer aims to generate. Hence, this statement is exclusively dependent on the properties of the GNN classifier $f(.)$ than on any specific dataset.  Further, this statement is validated in Table 2 on different target classes of various datasets where the obtained class score across 50 explanations is 1. As demonstrated by the theoretical guarantee in Appendix D.2, it is highly improbable that two graphs are same in this set of generated graphs. We hope this clarifies what we meant by the statement.
>
> W2: Other works in GNN global-explanability are not cited [1][2][3][4].
>
> **Response**
>
> We thank the reviewer for pointing us to more literature on GNN Explainability, we have cited these papers in the Related Work section in the revised version.
>
> W3: Upon reviewing the experimental results (Figure 3b IMDB and reddit), it remains unclear what specific insights or explanations can be derived from the generated subgraph.
>
> **Response**
>
> We acknowledge that generated explanations for the IMDB and REDDIT datasets are larger than the explanations generated for other datasets. Note that both these datasets contain larger graphs( especially Reddit with average number of nodes being around 430 as reported in Table 1), hence it is difficult to faithfully capture the motifs identified by the classifier using smaller explanations. As discussed in our result section in detail, the generated explanations for REDDIT signify how the classifier understands the class identities based on the number of high degree nodes contained in the graph and for IMDB, the classifier understands class identities depending on how nodes are interconnected with each other in the graph. Since, the patterns have to do with high degree nodes and densely connected graphs, we thought it might not be possible to demonstrate this using smaller explanations(like in the case of other datasets) since it could not capture the inherent topological properties of the dataset due to the information bottleneck caused by its smaller size. However, we acknowledge that it might hinder the interpretability of the explanations due to which we have added in the Appendix I(Figure 9) a few smaller sized(not as small as the other datasets ) explanations, from which the inferences drawn in the discussion( Section 6.4) can still be concluded faithfully. We hope this addresses your concern.
>
> W4: It is unclear how node and edge features are handled by the graphon.
>
> **Response**
>
> We think our writing of handling the node and edge features might not have been clear enough in our manuscript. We address this in the revision by revising Algorithm 4 to include every detail of estimation of node features and sampling of graphs from the estimated graphon.

---

> > ### Author Response · Authors · 2024-09-11
> > **Reply to Reviewer 81Da (Continued)**
> >
> > **Requested Changes**
> >
> > 1) The authors should compare their method's performance against other relevant works.
> >
> > **Response**
> >
> > We would like to mention here that we have tried our best to conduct comprehensive experiments and analyse our explanations using various metrics quantitatively and qualitatively as noted by reviewers wLSB and 1V5r. It may be kindly noted that works [2] and [3] cannot be compared directly to our work since they approach explainability from a fundamentally different basis than our work. Authors of [2] propose a self-explainable GNN architecture which can explain its working mechanism on a dataset by generating a decision tree, hence their method  falls under the domain of intrinsic interpretability  rather than post-hoc interpretability of which model-level explanations are a sub-branch. Authors of [3] propose a method combining instance level and model level interpretability which takes explanations of an instance-level explainer as input and combine them using a logic to generate model-level explanations, hence the quality of these explanations would be dependent on the instance level explainer itself and we believe this does not yield itself to a fair comparison with our method which is a purely model-level explainer.     Papers [1] and [4] are indeed works on model-level explanations, however to the best of our knowledge we could not find a publicly available implementation of their approaches. We have supplemented experiments (Appendix J) to compare our method with [1] on three datasets, using two metrics, one of which is proposed by the authors of [1] themselves. We believe analysis of our results using this metric has added a new dimension to our paper, hence we thank the reviewer again for pointing us to this paper. However, since we could not find an implementation of their approach, we are forced to quote the results directly from their paper. We also could not compare with [4] since their approach is specifically designed for explaining working of a GNN classifier on molecular datasets and the datasets used in our paper and their paper do not overlap at all. We would also like to mention here that the notable advantage of our method over both these methods is the fact that our method can provide insights on the decision boundary of a classifier as well as on individual target classes while both of these methods have the restriction of generating explanations only for individual target classes.
> >
> > 2) Additionally, they should provide a more detailed discussion on the expected explanations in potential real-world applications. From an interpretability standpoint, it is difficult to discern meaningful insights from large graphs, such as those in Figure 3b (IMDB and Reddit).
> >
> > **Response**
> >
> > We thank the reviewer for identifying this pertinent point of discussing the application of our method in real life scenarios. We believe Graphon-Explainer can be used in many real world applications to inform an user about the inductive biases and underlying classification logic of a GNN model that is deployed in a real scenario. Like the experiment on MUTAG which demonstrates that the classifier looks at fused rings to identify mutagenic classes, our method can be potentially used to explain a GNN which classifies toxic and non-toxic drugs to determine the basis on which the GNN classifies a molecule as toxic. Analysis of decision boundary of such a GNN could provide insights on the robustness and the risk of misclassifying a particular toxic molecule as non-toxic. Also as demonstrated by the added experiments(in Appendix H), even in the scenario when two classifiers perform accurately their interpretations of a single class can differ greatly based on their inductive biases which can inform the user, which classifier to use even when two classifiers have equal accuracy in a real life scenario.  Further, our method can also generate explanations tailored to the query dataset an user possesses (since estimated Graphons can effectively learn the distribution of the dataset) which can provide tailored insights to an user depending on the kind of instances contained in the query dataset of each user rather than providing the same explanation to each user irrespective of the kind of instances they possess in their query dataset. We have also added this discussion in the Conclusion section of our revised paper.
> >
> > Yes, we hope we could clarify the need to have larger graphs as explanations for these two datasets in our answer to W3. You are requested to kindly refer to Figure 9 in the Appendix to find explanations containing smaller number of nodes.
> >
> > We would be happy to address any further concerns that you may have about our work.

---

> > > ### Author Response · Authors · 2024-09-19
> > > **Reply to Reviewer 81Da ( Added Comparative Results)**
> > >
> > > Dear Reviewer,
> > >
> > > Thank you again for your valuable suggestion of adding more comparative results. We have now incorporated a new baseline to compare our results for the generated motifs near the decision boundary. The current version of the manuscript contains the following comparative results:
> > > 1) Qualitative and quantitative comparison of generated explanations for target classes with XGNN and GNNInterpreter ( which was already present in the initial submission)
> > > 2)  **Newly Added:** Quantitative and qualitative comparison of generated decision boundary motifs with GNNBoundary (please see Table 3 in the main paper,  Figure 3(a&b) and Appendix K).
> > > 3) **Added in the previous revised version:** Comparison on three datasets (in Appendix J) with the method proposed in [1] using two metrics: class score and model fidelity ( as proposed by authors of [1]) .
> > >
> > > Best,
> > >
> > > Authors

---

### Author Response · Authors · 2024-09-26
**A Message for all Reviewers**

Dear Reviewers,

Thank you once again for your thoughtful and constructive feedback on our manuscript. We have tried to carefully address all of your major concerns through our point-by-point rebuttal, as well as in the revised manuscript, which we believe is significantly improved thanks to your input.

We would like to reiterate that the core objective of our approach is to develop an explainer that provides a comprehensive analysis of a classifier by examining both its understanding of class identities and its decision boundary properties. We believe that focusing on just one of these aspects offers an incomplete characterization. To the best of our knowledge, our method is the first to provide this level of model-level insight for a GNN classifier.

We kindly ask that the reviewers keep this central aim in mind when evaluating our contributions. We would also be happy to discuss any remaining questions or concerns you may have.

Best regards,

 Authors

---

### Author Response · Authors · 2024-10-17
**Any further questions following our Rebuttal and Revision**

Dear Reviewers,

Thank you again for your thorough and constructive feedback of our manuscript. Please feel free to let us know if you have any additional questions, comments or concerns about our manuscript following our rebuttal and the latest revision.

Thank You

---

### Decision · Action_Editor_7wFx · 2024-10-24

**Recommendation:** Accept as is

**Comment:**

All the reviewers agree that the paper is well written and includes extensive experiments with helpful illustrations. There are two concerns that were not addressed according to the reviewers:
1. Explanations provided for the larger graphs is unclear. It is helpful if authors can improve this part in the camera-ready version.
2.  Whether the explanations generated with the method is informative or useful for the user.  However, the reviewer also points out that this is a subjective point.

**Audience:**

Yes, the presented paper is of interest to GNN community.

**Claims And Evidence:**

There was a consensus among the reviewers that the claims made by the authors are supported. The authors addressed most of the concerns raised by the reviewers and the reviewers were satisfied with the additional information and experiments provided by the authors.

---

> ### Author Response · Authors · 2024-10-29
> **Thank You!**
>
> We thank the AE for coordinating what was a very fast and satisfying reviewing process. We are very happy that the reviewers reached a consensus to accept our paper. We thank all the reviewers and AE again for a very thorough and timely review of our paper and deciding in favor of accepting the paper. We are currently preparing the camera ready version and we will make sure that we incorporate the concerns that all the reviewers raised and the recommended changes by the AE.
>
>
> Best Wishes,
>
> Authors

---

> ### Author Response · Authors · 2024-11-01
> **Added Camera Ready Version**
>
> Dear AE,
>
> We have added the camera ready version, incorporating both of your suggestions. Please let us know if any further adjustments are needed. We thank you again for your time and consideration.
>
> Best Wishes,
>
> Authors